# Towards Escaping from Class Dependency Modeling for Multi-Dimensional Classification

**Teng Huang** [1 2]  **Bin-Bin Jia** [3]  **Min-Ling Zhang** [1 2]

## Abstract

In multi-dimensional classification (MDC), the semantics of objects are characterized by multiple class variables from different dimensions. Existing MDC approaches focus on designing effective class dependency modeling strategies to enhance classification performance. However, the inter-coupling of multiple class variables poses a significant challenge to the precise modeling of class dependencies. In this paper, we make the first attempt towards escaping from class dependency modeling for addressing MDC problems. Accordingly, a novel MDC approach named DCOM is proposed by decoupling the interactions of different dimensions in MDC. Specifically, DCOM endeavors to identify a latent factor that encapsulates the most salient and critical feature information. This factor will facilitate partial conditional independence among class variables conditioned on both the original feature vector and the learned latent embedding. Once the conditional independence is established, classification models can be readily induced by employing simple neural networks on each dimension. Extensive experiments conducted on benchmark data sets demonstrate that DCOM outperforms other state-of-the-art MDC approaches.

## 1. Introduction

In practical applications, objects are always characterized by diverse semantics across multiple dimensions. To achieve a more nuanced and comprehensive depiction of these objects, multi-dimensional classification (MDC) focuses on the analysis of multiple class spaces carrying distinct semantics aspects. For example, Figure 1 (Liu et al., 2016) illustrates a cloth image characterized by three dimensions including `Texture`, `style` and `Elasticity`, each with corresponding possible labels. In reality, the necessity for learning from MDC objects is prevalent across a variety of real-world applications, including computer vision (Lian et al., 2020; Shi et al., 2025), text mining (Lertnattee & Theeramunkong, 2004; Serafino et al., 2015), ecological informatics (Dzeroski et al., 2000; Verma et al., 2021), etc.

Formally speaking, let $\mathcal{X} = \mathbb{R}^d$ be the input (feature) space and $\mathcal{Y} = C_1 \times C_2 \times \cdots \times C_q$ be the output space which corresponds to the Cartesian product of $q$ class spaces. Each class space $C_j = \{c_1^j, c_2^j, \ldots, c_{K_j}^j\}$ consists of $K_j$ possible class labels ($1 \le j \le q$). In addition, denote the $d$-dimensional feature variable by $X$ defined on $\mathcal{X}$ and the associated $q$-dimensional class variable by $Y = (Y_1, Y_2, \ldots, Y_q)$ defined on $\mathcal{Y}$, where each component $Y_j$ is a scalar class variable defined on class space $C_j$ ($1 \le j \le q$). Given an MDC data set $\mathcal{D} = \{(\boldsymbol{x}_i, \boldsymbol{y}_i) | 1 \le i \le m\}$ consisting of $m$ i.i.d. training examples sampled from the Cartesian product of input space and output space $\mathcal{X} \times \mathcal{Y}$, the aim of discriminative MDC approaches is to estimate the conditional joint probability $p_{Y|X}(\boldsymbol{y}_\diamond | \boldsymbol{x}_\diamond) = p_{Y_1, Y_2 \ldots, Y_q | X}(y_{\diamond 1}, y_{\diamond 2} \ldots, y_{\diamond q} | \boldsymbol{x}_\diamond)$[1] by training on the data set $\mathcal{D}$.

Specifically, for an unseen sample $\boldsymbol{x}_*$, the predicted class vector $\hat{\boldsymbol{y}}_*$ is exactly $\arg\max_{\boldsymbol{y}_\diamond \in \mathcal{Y}} p_{Y|X}(\boldsymbol{y}_\diamond | \boldsymbol{x}_\diamond = \boldsymbol{x}_*)$. One straightforward strategy is to transform the multi-variant class vector $\boldsymbol{y}_\diamond$ into a scalar variable $y^{(\text{cp})}$ with a predefined bijective mapping function $\phi : \mathcal{Y} \to \{1, 2, \ldots, K_1 \cdot K_2 \cdot \ldots \cdot K_j\}$. In other words, each distinct class combination is regarded as a new class and multi-class classification algorithms can be utilized subsequently, which is known as class powerset (CP) (Read et al., 2014b). Then the targeted probability is transformed into $p_{Y|X}(\phi(y_{\diamond 1}, \ldots, y_{\diamond q}) | \boldsymbol{x}_\diamond)$. However, the considered sample space is still the whole output space $\mathcal{Y}$, which contains a tremendous number of elements (i.e., the cardinality of $\mathcal{Y}$, denoted by $|\mathcal{Y}| = \prod_{j=1}^q K_j$) and

---

[1]School of Computer Science and Engineering, Southeast University, Nanjing 210096, China [2]Key Lab. of Computer Network and Information Integration (Southeast University), MOE, China [3]College of Electrical and Information Engineering, Lanzhou University of Technology, Lanzhou 730050, China. Correspondence to: Min-Ling Zhang <zhangml@seu.edu.cn>.

*Proceedings of the $42^{nd}$ International Conference on Machine Learning*, Vancouver, Canada. PMLR 267, 2025. Copyright 2025 by the author(s).

[1]Note that $\boldsymbol{x}_\diamond$ is a randomly sampled value of the feature variable $X$ and $\boldsymbol{x}_i$ represents the $i$-th feature vector in the data set $\mathcal{D}$. Similarly, $\boldsymbol{y}_\diamond, y_{\diamond 1}, \ldots, y_{\diamond q}$ are randomly sampled values of class variables $Y, Y_1, \ldots, Y_q$ and $\boldsymbol{y}_i$ represents the $i$-th class vector in the data set $\mathcal{D}$. Additionally, $y_{ij}$ denotes the $j$-th elements of $\boldsymbol{y}_i$.

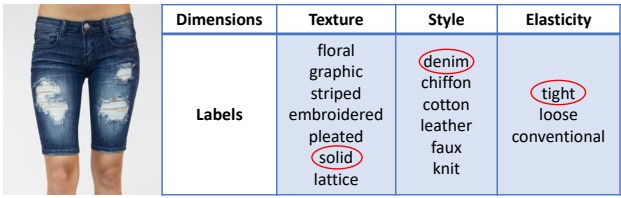

| Dimensions | Texture | Style | Elasticity |
| --- | --- | --- | --- |
| Labels | floral graphic striped embroidered pleated (solid) lattice | (denim) chiffon cotton leather faux knit | (tight) loose conventional |

*Figure 1.* An image example of multi-dimensional classification. Labels in each column on a blue background belong to the corresponding dimension in the first row. The ground truth labels *solid*, *denim* and *tight* are in red circles.

is prone to class-imbalance and overfitting problem. Another natural strategy is to focus on each class space independently so that the conditional joint probability $p_{Y|X}(\boldsymbol{y}_\diamond|\boldsymbol{x}_\diamond)$ is transformed to $q$ independent conditional marginal probability $p_{Y_j|X}(y_{\diamond j}|\boldsymbol{x}_\diamond)$ $(1 \le j \le q)$, known as binary relevance (BR) (Zhang et al., 2018). Although BR significantly reduces the cardinality of probability space to be measured, the class dependencies among dimensions are thoroughly ignored, which may result in suboptimal prediction performance. This highlights a critical need for exploring an accurate approach with theoretical interpretation to model class dependencies.

Accordingly, existing MDC approaches generally dedicate to appropriately modeling class dependencies as well as avoiding confronting the vast original output space $\mathcal{Y}$ directly. On the one hand, some approaches opt to account for local class dependencies among a subset of class variables, rather than the entire set, such as probabilistic graph-based approaches (Bielza et al., 2011; Gil-Begue et al., 2021; Nguyen et al., 2023), super-class modeling approaches (Read et al., 2014a) and pairwise dependency modeling approaches (Jia & Zhang, 2020; Huang et al., 2024). On the other hand, alternative approaches are designed to transform the original input or output space into novel representations that facilitate more implicit dependency modeling processes (Ma & Chen, 2018; Wang et al., 2020; Jia & Zhang, 2023).

However, modeling dependencies in MDC remains a significant challenge. Based on the above probabilistic analysis on BR and CP, the hardness of modeling class dependencies stems from the typical intercoupling within multiple dimensions. This multi-dimensional intercoupling arises from the conditional dependency among class variables given the feature variable, namely the discrepancy between the conditional joint probability $p_{Y|X}(\boldsymbol{y}_\diamond|\boldsymbol{x}_\diamond)$ and the product of conditional marginal probability $\prod_{1 \le j \le q} p_{Y_j|X}(y_{\diamond j}|\boldsymbol{x}_\diamond)$. As a matter of fact, given an MDC instance, it is highly improbable that these two forms of probability are identical, unless there is a complete absence of correlation among dimen-

sions involved. To address this intricate class dependency modeling problem from a dual perspective, we propose a novel MDC approach named DCOM (i.e., *DeCOupling Multi-dimensional classification*). Instead of modeling class dependencies directly, we attempt to identify a latent factor that encapsulates the most essential and critical feature information, which enables partial conditional independence among class variables conditioned on both the original feature variable and the learned latent variable. Furthermore, once the conditional independence is established, classification models can be induced easily by employing simple neural networks on each dimension. The main contributions of DCOM are summarized as follows:

- We make a first attempt towards escaping from class dependency modeling in MDC. The proposed decoupling strategy can also be generalized to other multi-output learning paradigms, such as multi-task learning (Feng & Chen, 2023) and multi-label classification (Shi et al., 2024; Zhang & Zhang, 2024; Sun et al., 2024).

- From a probabilistic standpoint, we present an efficient approximation method. This method fosters the conditional independence among class variables conditioned on the informative latent factor and feature vector, accompanied by a theoretical analysis for dealing with the vast sample space.

- Comprehensive experiments over seventeen benchmark data sets demonstrate that DCOM outperforms other state-of-the-art MDC approaches.

The rest of this paper is organized as follows. Firstly, related works are briefly reviewed in Section 2. Secondly, the details of the proposed DCOM approach are presented in Section 3. Thirdly, experimental results of comparative studies on benchmark multi-dimensional data sets are reported in Section 4. Finally, Section 5 concludes this paper.

## 2. Related Work

Most existing MDC approaches dedicate to modeling the class dependencies as well as avoiding confronting the huge original output space $\mathcal{Y}$ directly. Amongst them, chain-based models (Zaragoza et al., 2011; Read et al., 2014b) disassembles the conditional joint distribution into the product of $q$ distributions by the chain rule of probability, i.e., $p_{Y_1,Y_2,...,Y_q|X}(y_{\diamond 1}, y_{\diamond 2}, \ldots, y_{\diamond q}|\boldsymbol{x}_\diamond) = \prod_{j=1}^{q} p_{Y_j|X,Y_1,...,Y_{j-1}}(y_{\diamond j}|\boldsymbol{x}_\diamond, y_{\diamond 1}, \ldots, y_{\diamond(j-1)})$ and estimates each individual probability term with a multi-class classifier. Such step-by-step strategy skillfully makes learning on each classifier achievable by considering single class space on each step but leads to the propagation and accumulation of errors (Read et al., 2014b). Probabilistic graph models (Bielza et al., 2011; Gil-Begue et al., 2021;

Nguyen et al., 2023) seek for potential relationship among class variables by directed acyclic graphs (DAG). Thus the global joint distribution can be partitioned into some local joint distributions involving class variables and their respective parent variables. Nevertheless, the computational complexity associated with determining DAG structure remains substantial. Dimension-specific MDC approach (Huang et al., 2024) transforms the original feature vector $\boldsymbol{x}$ into dimension-specific feature vectors $\boldsymbol{d}^{(j)}$ corresponding to the $j$-th dimension. Then the modeling problem is transformed into $p_{Y_j|D}(y_{\diamond j}|\boldsymbol{d}^{(j)})$ with each class space being treated as an individual sample space.

Another category of MDC approaches aim to transform the original input or output space into new representations, thereby facilitating implicit dependency modeling processes. gMML (Ma & Chen, 2018) decomposes the class spaces into a binary-valued label space via one-vs-rest strategy and solves the resulting problem via a metric approach. LEFA (Wang et al., 2020) learns latent label embeddings based on attentional factorization machines to augment the original feature space. ADVAE-FLOW (Zhang et al., 2022) encodes both feature and class variables into probabilistic latent spaces by normalizing flows. DLEM (Jia & Zhang, 2023) enables modeling alignment in an encoded label space derived from one-vs-one decomposition and transforms the output space into a ternary encoded label space.

However, no existing MDC approaches attempt to escape from the complicated dependency modeling problem. In the next section, we will elaborate the technical details of the proposed DCOM approach, which seeks for partial conditional independence based on the instance-based conditional mutual information.

## 3. The DCOM Approach

**Notation.** In this paper, we use capital letters to denote random variables (e.g., $Y, X, Z$), lowercase bold letters to denote vectors (e.g., $\boldsymbol{y}_\diamond, \boldsymbol{x}_i$) and lowercase non-bold letters to denote scalars (e.g., $y_{\diamond j}, y_{ij}$). A detailed summary of notations can be found in Table 8 of Appendix C.2.

### 3.1. Latent Factor Introduction

Given the MDC training data set $\mathcal{D} = \{(\boldsymbol{x}_i, \boldsymbol{y}_i)|1 \leq i \leq m\}$ consisting of $m$ i.i.d. training examples sampled from $\mathcal{X} \times \mathcal{Y}$. The conditional log-likelihood function in terms of class variables conditioned on the feature variable can be given as follows:

$$
\begin{aligned}
L_0(\boldsymbol{\Theta}_0) &= \sum_{i=1}^{m} \log p_{Y|X}(\boldsymbol{y}_i|\boldsymbol{x}_i; \boldsymbol{\Theta}_0) \\
&= \sum_{i=1}^{m} \log p_{Y|X}(y_{i1}, \ldots, y_{iq}|\boldsymbol{x}_i; \boldsymbol{\Theta}_0), \quad (1)
\end{aligned}
$$

where $\boldsymbol{\Theta}_0$ is the model parameters. Recall that for the $j$-th class variable $Y_j$, there exist $K_j$ possible class labels $c_1^j, c_2^j, \ldots, c_{K_j}^j$. Consequently, the cardinality of the probability space is given by $|\mathcal{Y}| = \prod_{j=1}^{q} K_j$, which is exceedingly large and poses a significant challenge for precise estimation. Considering the equivalence between maximizing likelihood estimation (MLE) and minimizing the cross entropy loss, the loss item regarding classification in existing deep MDC approaches (Huang et al., 2024; Shi et al., 2025) is equivalent to the summation of conditional marginal log-likelihood function defined as follows:

$$
L_1(\boldsymbol{\Theta}_1) = \sum_{i=1}^{m} \sum_{j=1}^{q} \log p_{Y_j|X}(y_{ij}|\boldsymbol{x}_i; \boldsymbol{\theta}^{(j)}), \quad (2)
$$

where $\boldsymbol{\Theta}_1 = [\boldsymbol{\theta}^{(1)}, \boldsymbol{\theta}^{(2)}, \ldots, \boldsymbol{\theta}^{(q)}]$ is the set of model parameters and $\boldsymbol{\theta}^{(j)}$ represents the model parameter of the $j$-th dimension. Here, the predictive model of the $j$-th dimension only considers the $j$-th class space $C_j$ and the total cardinality of probability space is $\sum_{j=1}^{q} K_j$, which is much less than $\prod_{j=1}^{q} K_j$. However, the equality of Eq.(1) and Eq.(2) requires an implicit assumption on the partial[2] conditional independence among class variables conditioned on the feature variable, which is formulated as:

$$
p_{Y|X}(\boldsymbol{y}_i|\boldsymbol{x_i}) = \prod_{j=1}^{q} p_{Y_j|X}(y_{ij}|\boldsymbol{x}_i), \quad (3)
$$

where $i \in \{1, 2, \ldots, m\}$. Unfortunately in the context of MDC, this strong assumption is rarely tenable. To facilitate the validity of the partial conditional independence and thereby reduce the cardinality of the probability space, DCOM seeks to identify a high-level informative latent factor denoted by variable $Z$ that enhances the original and basic feature information. Specifically, we assume that the prior over the latent variable distribution is the centered isotropic multivariate Gaussian $p_Z(\boldsymbol{z}_\diamond) = \mathcal{N}(\boldsymbol{0}, \mathbf{I})$. We then employ a simple encoding network $\mathcal{G}$ to derive the latent vector $\boldsymbol{z}_\diamond$ corresponding to each feature vector $\boldsymbol{x}_\diamond$. Nevertheless, such a direct mapping from $\mathcal{X}$ is incapable of altering the conditional dependence since the conditional probability $p(\mathcal{G}(\boldsymbol{x}_\diamond)|\boldsymbol{x}_\diamond)$ is inherently describing a deterministic event. To address this issue, we introduce a minor perturbation on $\boldsymbol{x}_\diamond$ prior to applying the encoding network $\mathcal{G}$. Consequently, the latent vector can be computed as follows:

$$
\boldsymbol{z}_\diamond = \mathcal{G}(\tilde{\boldsymbol{x}}_\diamond), \quad (4)
$$

where $\tilde{\boldsymbol{x}}_\diamond = \boldsymbol{x}_\diamond + \boldsymbol{\epsilon}$ and $\boldsymbol{\epsilon}$ is a random noise vector. Furthermore, in order to comprehensively assess the influence of the

---

[2]We use "partial" here because the assumption does not require Eq.(3) to hold universally for all possible values of $Y$ and $X$, but rather only under the observed training distribution (which is finite and empirically sampled).

latent variable $Z$ on both feature variable $X$ and class variable $Y$, we extend the original conditional log-likelihood function conditioned on $X$ (i.e., Eq.(1)) to a conditional log-likelihood function conditioned on $Z$ as follows:

$$
\begin{aligned}
L(\mathbf{\Theta}) &= \sum_{i=1}^{m} \log p_{Y,X|Z}(\boldsymbol{y}_i, \boldsymbol{x}_i | \boldsymbol{z}_i; \mathbf{\Theta}) \\
&= \sum_{i=1}^{m} \Big[ \log p_{Y|X,Z}(\boldsymbol{y}_i | \boldsymbol{x}_i, \boldsymbol{z}_i; \boldsymbol{\theta}_d) \\
&\quad + \log p_{X|Z}(\boldsymbol{x}_i | \boldsymbol{z}_i; \boldsymbol{\theta}_r) \Big],
\end{aligned} \tag{5}
$$

where $\mathbf{\Theta} = [\boldsymbol{\theta}_d, \boldsymbol{\theta}_r]$ is the set of model parameters. $\boldsymbol{\theta}_d$ and $\boldsymbol{\theta}_r$ correspond to the parameters of the discriminant component and reconstruction component, respectively.

### 3.2. Conditional Independence Achievement

Although the challenge posed by the vast probability space associated with the first term in Eq.(5) persists, the introduction of the latent variable $Z$ offers the potential for achieving the partial conditional independence among class variables conditioned on the feature variable $X$ as well as the latent variable $Z$. The desired partial conditional independence is formulated as

$$
p_{Y|X,Z}(\boldsymbol{y_i}|\boldsymbol{x_i}, \boldsymbol{z_i}) = \prod_{j=1}^{q} p_{Y_j|X,Z}(y_{ij}|\boldsymbol{x_i}, \boldsymbol{z_i}), \tag{6}
$$

where $i \in \{1, 2, \ldots, m\}$. For notational conciseness, we further denote the joint probability of all variables $p_{Y,X,Z}(\boldsymbol{y}_\diamond, \boldsymbol{x}_\diamond, \boldsymbol{z}_\diamond)$ by $p^{(\mathrm{jt})}(\boldsymbol{y}_\diamond, \boldsymbol{x}_\diamond, \boldsymbol{z}_\diamond)$ and the conditional marginal probability multiplied by the joint probability of feature and latent variables $\prod_{j=1}^{q} p_{Y_j|X,Z}(y_{\diamond j}|\boldsymbol{x}_\diamond, \boldsymbol{z}_\diamond) p_{X,Z}(\boldsymbol{x}_\diamond, \boldsymbol{z}_\diamond)$ as $p^{(\mathrm{pd})}(\boldsymbol{y}_\diamond, \boldsymbol{x}_\diamond, \boldsymbol{z}_\diamond)$.[3] Then Eq.(6) is equivalent to:

$$
p^{(\mathrm{jt})}(\boldsymbol{y}_i, \boldsymbol{x}_i, \boldsymbol{z}_i) = p^{(\mathrm{pd})}(\boldsymbol{y}_i, \boldsymbol{x}_i, \boldsymbol{z}_i). \tag{7}
$$

To provide a theoretical support, we define the distance between the two sides of Eq.(7), i.e., $p^{(\mathrm{jt})}$ and $p^{(\mathrm{pd})}$, inspired by Kullback-Leibler (KL) divergence and conditional mutual information as follows:

**Definition 3.1.** Given $q$ class variables $Y_1, Y_2, \ldots, Y_q$ defined on $\mathcal{Y} = C_1 \times C_2 \times \ldots C_q$, a feature variable $X$ defined on $\mathcal{X}$ and a latent variable $Z \sim \mathcal{N}(\mathbf{0}, \mathbf{I})$, the *conditional*

---

[3]When no ambiguity arises, we abbreviate $p^{(\mathrm{jt})}(\boldsymbol{y}_\diamond, \boldsymbol{x}_\diamond, \boldsymbol{z}_\diamond)$ as $p^{(\mathrm{jt})}$ and $p^{(\mathrm{pd})}(\boldsymbol{y}_\diamond, \boldsymbol{x}_\diamond, \boldsymbol{z}_\diamond)$ as $p^{(\mathrm{pd})}$.

*mutual information* is defined as:

$$
\begin{aligned}
& I(Y_1, Y_2, \ldots, Y_q | X, Z) \\
=& \ KL\left(p^{(\mathrm{jt})} || p^{(\mathrm{pd})}\right) \\
=& \sum_{\boldsymbol{y}_\diamond \in \mathcal{Y}} \int_{\boldsymbol{x}_\diamond} \int_{\boldsymbol{z}_\diamond} p^{(\mathrm{jt})} \log \frac{p^{(\mathrm{jt})}}{p^{(\mathrm{pd})}} d\boldsymbol{z}_\diamond \\
=& \sum_{\boldsymbol{y}_\diamond \in \mathcal{Y}} \int_{\boldsymbol{x}_\diamond} \int_{\boldsymbol{z}_\diamond} \mathcal{I}(\boldsymbol{y}_\diamond, \boldsymbol{x}_\diamond, \boldsymbol{z}_\diamond) d\boldsymbol{z}_\diamond. \tag{8}
\end{aligned}
$$

For convenience, the above integrand has been denoted as:

$$
\mathcal{I}(\boldsymbol{y}_\diamond, \boldsymbol{x}_\diamond, \boldsymbol{z}_\diamond) = p^{(\mathrm{jt})}(\boldsymbol{y}_\diamond, \boldsymbol{x}_\diamond, \boldsymbol{z}_\diamond) \log \frac{p^{(\mathrm{jt})}(\boldsymbol{y}_\diamond, \boldsymbol{x}_\diamond, \boldsymbol{z}_\diamond)}{p^{(\mathrm{pd})}(\boldsymbol{y}_\diamond, \boldsymbol{x}_\diamond, \boldsymbol{z}_\diamond)}.
$$

Given the non-negativeness of KL divergence, it is evident that the condition wherein Eq.(8) equals 0 constitutes a sufficient but not necessary condition for the satisfaction of Eq.(7) since a multitude of possible values of the feature and latent variable are intractable and not considered in Eq.(7). Therefore, for a given MDC data set, we employ an instance-based conditional mutual information defined as follows:

**Definition 3.2.** For a given MDC data set $\mathcal{D} = \{(\boldsymbol{x}_i, \boldsymbol{y}_i)|1 \leq i \leq m\}$, the *instance-based conditional mutual information* is defined as:

$$
I_{\mathcal{D}}(Y_1, Y_2, \ldots, Y_q | X, Z) = \sum_{i=1}^{m} \sum_{\boldsymbol{y}_\diamond \in \mathcal{Y}} \mathcal{I}(\boldsymbol{y}_\diamond, \boldsymbol{x}_i, \boldsymbol{z}_i). \tag{9}
$$

In this paper, we show a feasible and efficient way to estimate the instance-based conditional mutual information with Theorem 3.3.

**Theorem 3.3.** *If the joint probability of class variables $p_Y(y_{\diamond 1}, y_{\diamond 2}, \ldots, y_{\diamond q})$ is small enough, i.e., $0 \leq p_Y(y_{\diamond 1}, y_{\diamond 2}, \ldots, y_{\diamond q}) \ll 1$, for $\forall \delta \in \mathbb{R}$ s.t. $0 \leq \delta \leq p^{(\mathrm{jt})}(\boldsymbol{y}_\diamond, \boldsymbol{x}_\diamond, \boldsymbol{z}_\diamond)$, then we have $|\mathcal{I}(\boldsymbol{y}_\diamond, \boldsymbol{x}_\diamond, \boldsymbol{z}_\diamond)| \to 0$ under Assumption 3.4.*

**Assumption 3.4.** Given $\forall \delta \in \mathbb{R}$, s.t. $0 \leq \delta \leq p^{(\mathrm{jt})}(\boldsymbol{y}_\diamond, \boldsymbol{x}_\diamond, \boldsymbol{z}_\diamond)$, then the joint probability $p^{(\mathrm{jt})}(\boldsymbol{y}_\diamond, \boldsymbol{x}_\diamond, \boldsymbol{z}_\diamond)$ and the product of conditional marginal probability $p^{(\mathrm{pd})}(\boldsymbol{y}_\diamond, \boldsymbol{x}_\diamond, \boldsymbol{z}_\diamond)$ satisfy that $p^{(\mathrm{jt})}(\boldsymbol{y}_\diamond, \boldsymbol{x}_\diamond, \boldsymbol{z}_\diamond) - \delta \leq p^{(\mathrm{pd})}(\boldsymbol{y}_\diamond, \boldsymbol{x}_\diamond, \boldsymbol{z}_\diamond)$.

As a matter of fact, the joint probability $p^{(\mathrm{jt})}(\boldsymbol{y}_\diamond, \boldsymbol{x}_\diamond, \boldsymbol{z}_\diamond) \leq \int_{\boldsymbol{x}_\diamond} \int_{\boldsymbol{z}_\diamond} p^{(\mathrm{jt})}(\boldsymbol{y}_\diamond, \boldsymbol{x}_\diamond, \boldsymbol{z}_\diamond) d\boldsymbol{z}_\diamond \leq p_Y(y_{\diamond 1}, \ldots, y_{\diamond q}) \to 0$ when the condition of Theorem 3.3 is satisfied. Consequently, $p^{(\mathrm{jt})} - \delta$ is also a small probability that tends to 0. This observation suggests that the Assumption 3.4 is highly likely to be valid. Theorem 3.3 provides an efficient strategy to reduce the practical size of probability space considering a specific MDC data set, which shows that when the prior

joint probability of class variables $p_Y(y_{\diamond 1}, \ldots, y_{\diamond q})$ is small enough, the corresponding summand in Eq.(9) can be disregarded. We defer the proof of Theorem 3.3 to Appendix A. Moreover, this prior which only regards to class variables can be computed easily based on the $m$ class vectors of the i.i.d. MDC data set. Specifically, given an MDC data set $\mathcal{D} = \{(\boldsymbol{x}_i, \boldsymbol{y}_i) | 1 \leq i \leq m\}$, we can approximate the instance-based conditional mutual information as follows:

$$I_{\mathcal{D}'}(Y_1, Y_2, \ldots, Y_q | X, Z) = \sum_{i=1}^{m} \sum_{\boldsymbol{y}_\diamond \in \mathcal{D}'} \mathcal{I}(\boldsymbol{y}_\diamond, \boldsymbol{x}_i, \boldsymbol{z}_i). \quad (10)$$

Here, $\mathcal{D}'$ is defined as

$$\mathcal{D}' = \{\boldsymbol{y}_\diamond | \frac{\#\boldsymbol{y}_\diamond}{m} > c, \boldsymbol{y}_\diamond \in \mathcal{D}\},$$

where $\#\boldsymbol{y}_\diamond$ indicates the count of class vectors in data set $\mathcal{D}$ that are equal to $\boldsymbol{y}_\diamond$ and $c$ is a small constant that approaches 0. The transformation on the sample space of class variables considered from $\mathcal{Y}$ into $\mathcal{D}'$ makes the instance-based conditional independence computable. In real-world MDC data sets, the cardinality of $\mathcal{D}'$ is typically deemed acceptable since the inherent class dependencies across dimensions necessitate the presence of highly correlated class vectors. Now, the intractable conditional joint log-likelihood (i.e., Eq.(5)) can be transformed into an accessible form with the minimization of Eq.(10) as follows:

$$\tilde{L}(\tilde{\boldsymbol{\Theta}}) = \sum_{i=1}^{m} \Big[ \sum_{j=1}^{q} \log p_{Y_j | X, Z}(y_{ij} | \boldsymbol{x}_i, \boldsymbol{z}_i; \boldsymbol{\theta}_d^{(j)})$$
$$+ \log p_{X | Z}(\boldsymbol{x}_i | \boldsymbol{z}_i; \boldsymbol{\theta}_r) \Big], \quad (11)$$

where $\tilde{\boldsymbol{\Theta}} = [\boldsymbol{\theta}_d^{(1)}, \boldsymbol{\theta}_d^{(2)}, \ldots, \boldsymbol{\theta}_d^{(q)}, \boldsymbol{\theta}_r]$ is the set of model parameters. $\boldsymbol{\theta}_d^{(j)}$ and $\boldsymbol{\theta}_r$ correspond to the parameters of the discriminant component for the $j$-th dimension ($1 \leq j \leq q$) and the reconstruction component, respectively. The detailed construction of loss functions based on Eq.(11) will be discussed in the next section.

### 3.3. Loss Function Construction

For the first term of Eq.(11) regarding the discriminant component consisting of the $q$ conditional probabilities $\{p_{Y_j | X, Z}(y_{\diamond j} | \boldsymbol{x}_\diamond, \boldsymbol{z}_\diamond) | 1 \leq j \leq q\}$, we utilize $q$ independent neural networks $\{\mathcal{H}_j | 1 \leq j \leq q\}$ to model the corresponding conditional probability. Specifically, considering the $i$-th training example, for the $j$-th dimension, the input of $\mathcal{H}_j$ will be set as the concatenation of $\boldsymbol{x}_i$ and $\boldsymbol{z}_i$ and the output of $\mathcal{H}_j$ is transformed to normalized probabilities as follows:

$$\begin{aligned} & p_{Y_j | X, Z}(y_{\diamond j} = c_a^j | \boldsymbol{x}_i, \boldsymbol{z}_i) \\ = & [\zeta(\mathcal{H}_j([\boldsymbol{x}_i, \boldsymbol{z}_i]))]_a \\ = & \frac{\exp(\mathcal{H}_j([\boldsymbol{x}_i, \boldsymbol{z}_i])_a)}{\sum_{b=1}^{K_j} \exp(\mathcal{H}_j([\boldsymbol{x}_i, \boldsymbol{z}_i])_b)}, \end{aligned} \quad (12)$$

where $\zeta$ denotes the soft-max function and the subscript $a$ and $b$ denote the $a$-th and $b$-th element of the corresponding output vectors, respectively.

For the second term of Eq.(11) regarding the reconstruction component, we assume $p_{X | Z}(\boldsymbol{x}_i | \boldsymbol{z}_i)$ is a multivariate Gaussian with a diagonal covariance structure $\mathcal{N}(\boldsymbol{\mu}_i, \boldsymbol{\sigma}_i^2 \mathbf{I})$. Here, $\boldsymbol{\mu}_i$ and $\boldsymbol{\sigma}_i$ are the $d$-dimensional mean and the standard deviation vectors, which are the output of a reconstruction network $\mathcal{R}$. Then for the $i$-th training example, the logarithmic posterior probability can be computed as:

$$\begin{aligned} \log p_{X | Z}(\boldsymbol{x}_i | \boldsymbol{z}_i) = & -\frac{d}{2} \log 2\pi - \frac{1}{2} \sum_{a=1}^{d} \log \sigma_{ia}^2 \\ & -\frac{1}{2} \sum_{a=1}^{d} \frac{(x_{ia} - \mu_{ia})^2}{\sigma_{ia}^2}, \end{aligned} \quad (13)$$

where $d$ and $x_{ia}$ denotes the dimensionality and the $a$-th element of $\boldsymbol{x}_i$, respectively. To simplify the model, it is common to assume that the variances $\{\sigma_{ia}^2 | 1 \leq a \leq d\}$ are identical and constant. Under this assumption, the reconstruction network $\mathcal{R}$ only needs to output the mean parameters $\boldsymbol{\mu}_i$. Then the reconstruction loss can be computed as follows:

$$\mathcal{L}_{re} = \sum_{i=1}^{m} \sum_{a=1}^{d} -\frac{(x_{ia} - \mu_{ia})^2}{2}. \quad (14)$$

For the approximated instance-based conditional mutual information, i.e., Eq.(10), we need to estimate $p^{(\text{jt})}$ and $p^{(\text{pd})}$. According to the chain rule of probability, $p^{(\text{jt})}$ can be decomposed as $p_Z(\boldsymbol{z}_i) p_{X | Z}(\boldsymbol{x}_i | \boldsymbol{z}_i) p_{Y | X, Z}(\boldsymbol{y}_\diamond | \boldsymbol{x}_i, \boldsymbol{z}_i)$, where the first term can be computed as $p_Z(\boldsymbol{z}_i) = \mathcal{N}(\boldsymbol{z}_i; \mathbf{0}, \mathbf{I})$ following the normal distribution and the second term $p_{X | Z}(\boldsymbol{x}_i | \boldsymbol{z}_i)$ can be computed as Eq.(13) without the logarithmic function. As for the last term $p_{Y | X, Z}(\boldsymbol{y}_\diamond | \boldsymbol{x}_i, \boldsymbol{z}_i)$, we employ a neural network $\mathcal{T}$ with $|\mathcal{D}'|$ output nodes to estimate this probability with soft-max function as Eq.(12). Then the classification loss regarding classification networks $\{\mathcal{H}_j | 1 \leq j \leq q\}$ and $\mathcal{T}$ can be computed as the cross-entropy loss, formulated as:

$$\begin{aligned} \mathcal{L}_{ce} = & \sum_{i=1}^{m} \Big[ \log p_{Y | X, Z}(\boldsymbol{y}_i | \boldsymbol{x}_i, \boldsymbol{z}_i) \\ & + \sum_{j=1}^{q} \log p_{Y_j | X, Z}(y_{ij} | \boldsymbol{x}_i, \boldsymbol{z}_i) \Big]. \quad (15) \end{aligned}$$

Similarly, each term in $p^{(\text{pd})}(\boldsymbol{y}_i, \boldsymbol{x}_i, \boldsymbol{z}_i)$ can be estimated by existing neural networks. Therefore, the minimization of Eq.(10) can be regarded as the minimization of the following

loss function $\mathcal{L}_{ci}$, formulated as:

$$
\begin{aligned}
\mathcal{L}_{ci} \;=\; & \sum_{i=1}^{m} \exp\left(-\frac{\|\boldsymbol{x}_i - \boldsymbol{\mu}_i\|^2 + \|\boldsymbol{z}_i\|^2}{2}\right) \\
& \cdot \left[\sum_{\boldsymbol{y}_\diamond \in \mathcal{D}'} p_{Y|X,Z}(\boldsymbol{y}_\diamond|\boldsymbol{x}_i, \boldsymbol{z}_i) \right. \\
& \left. \cdot \log \frac{p_{Y|X,Z}(\boldsymbol{y}_\diamond|\boldsymbol{x}_i, \boldsymbol{z}_i)}{\prod_{j=1}^{q} p_{Y_j|X,Z}(y_{\diamond j}|\boldsymbol{x}_i, \boldsymbol{z}_i)}\right].(16)
\end{aligned}
$$

The overall loss function $\mathcal{L}$ is the combination of loss functions defined above:

$$
\mathcal{L} = \frac{1}{m}\left(\mathcal{L}_{ce} + \alpha \mathcal{L}_{re} + \beta \mathcal{L}_{ci}\right), \tag{17}
$$

where $\alpha$ and $\beta$ are trade-off parameters.

*Table 1.* Basic information for data sets. Here, $n, x$ and $image$ in last column represent numeric, nominal type and unstructured features.

| data set | #Exam. | #Dim. | #Labels/Dim. | #Feat. |
|---|---|---|---|---|
| Flare1 | 323 | 3 | 3,4,2 | $10x$ |
| Oes97 | 334 | 16 | 3 | $263n$ |
| Jura | 359 | 2 | 4,5 | $9n$ |
| Oes10 | 403 | 16 | 3 | $298n$ |
| Enb | 768 | 2 | 2,4 | $6n$ |
| Song | 785 | 3 | 3 | $98n$ |
| BeLaE | 1930 | 5 | 5 | $1n, 44x$ |
| Voice | 3136 | 2 | 4,2 | $19n$ |
| Thyroid | 9172 | 7 | 5,5,3,2,4,4,3 | $7n, 22x$ |
| CoIL2000 | 9822 | 5 | 6,10,10,4,2 | $81x$ |
| TIC2000 | 9822 | 3 | 6,4,2 | $83x$ |
| Flickr | 12198 | 5 | 3,4,3,4,4 | $1536n$ |
| Adult | 18419 | 4 | 7,7,5,2 | $5n, 5x$ |
| Default | 28779 | 4 | 2,7,4,2 | $14n, 6x$ |
| BP4D | 16037 | 7 | 6,6,6,6,6,2,8 | $image$ |
| DeepFashion | 20000 | 6 | 7,3,3,4,6,3 | $image$ |
| SEWA | 19275 | 3 | 21,20,19 | $image$ |

# 4. Experiments

## 4.1. Experimental Setting

### 4.1.1. DATA SETS

In this paper, we use seventeen real-world MDC data sets for experimental studies, including fourteen structured data sets[4] and three unstructured data sets: *BP4D* (Zhang

et al., 2013; 2014),[5] *DeepFashion* (Liu et al., 2016)[6] and *SEWA* (Kossaifi et al., 2021).[7] Table 1 summarizes basic characteristics, including the number of examples (#Exam.), the number of dimensions (#Dim.), the number of labels in each dimension (#Labels/Dim.) and the number of features (#Feat.).

### 4.1.2. EVALUATION METRICS

In this paper, we adopt three commonly used metrics for performance evaluation, i.e. *hamming score* (HS), *exact match* (EM) and *sub-exact match* (SEM) (Read et al., 2014a; Zhu et al., 2016). Given the test set $\mathcal{S} = \{(\boldsymbol{x}_i, \boldsymbol{y}_i) \mid 1 \le i \le p\}$ and the MDC model $f$ to be evaluated, the definitions of these three evaluation metrics are given as follows:

$$
\text{HS}_{\mathcal{S}}(f) = \frac{1}{p}\sum_{i=1}^{p}\frac{1}{q} \cdot r^{(i)}
$$

$$
\text{EM}_{\mathcal{S}}(f) = \frac{1}{p}\sum_{i=1}^{p}[\![r^{(i)} = q]\!]
$$

$$
\text{SEM}_{\mathcal{S}}(f) = \frac{1}{p}\sum_{i=1}^{p}[\![r^{(i)} \ge q - 1]\!].
$$

Here, $r^{(i)} = \sum_{j=1}^{q}[\![y_{ij} = \hat{y}_{ij}]\!]$ denotes the number of dimensions which are predicted correctly, where $y_{ij}$ and $\hat{y}_{ij}$ denote the ground-truth and predicted label w.r.t. the $j$-th dimension for the $i$-th test sample $\boldsymbol{x}_i$, respectively. $[\![\pi]\!]$ returns 1 if $\pi$ holds and 0 otherwise.

### 4.1.3. COMPARED APPROACHES

In this paper, the proposed DCOM approach is compared with seven state-of-the-art MDC approaches including BR, CP, DLEM, EDCC, LEFA, PIST and SLEM. Specifically, BR independently learns a multi-class classifier for each dimension, while CP learns a multi-class classifier via treating each distinct label combination as a new label. DLEM (Jia & Zhang, 2023) solves the MDC problem by enabling modeling alignment for MDC in an encoded label space derived from one-vs-one (OvO) decomposition. EDCC (Jia & Zhang, 2022) builds a chain of binary classifiers and augments the feature space by predictions generated by preceding classifiers. LEFA (Wang et al., 2020) introduces a cross correlation aware network to learn latent label embeddings and augments the original feature space by the learned label

---

[4] https://palm.seu.edu.cn/zhangml/Resources.htm#MDC_data

[5] https://www.cs.binghamton.edu/~lijun/Research/3DFE/3DFE_Analysis.html. See more detailed descriptions in Appendix B.

[6] https://mmlab.ie.cuhk.edu.hk/projects/DeepFashion/AttributePrediction.html. See more detailed descriptions in Appendix B.

[7] https://db.sewaproject.eu/. See more detailed descriptions in Appendix B

*Table 2.* Experimental results (mean±std.) of each MDC approach in terms of Hamming Score. In addition, ●/○ indicates whether DCOM is significantly superior/inferior to other compared approaches on each data set with pairwise t-test at 0.05 significance level.

| Data Set | DCOM | BR | CP | DLEM | EDCC | LEFA | PIST | SLEM |
|---|---|---|---|---|---|---|---|---|
| Flare1 | .920±.033 | .923±.031 | .881±.028● | .897±.035● | .923±.035 | .913±.034 | .923±.031 | .645±.262● |
| Oes97 | .749±.020 | .607±.031● | .686±.026● | .738±.026 | .747±.021 | .730±.024● | .737±.023● | .659±.016● |
| Jura | .740±.050 | .586±.065● | .675±.051● | .720±.057 | .616±.055● | .542±.059● | .602±.064● | .695±.066● |
| Oes10 | .813±.013 | .664±.018● | .746±.018● | .805±.015● | .797±.018● | .810±.008 | .806±.012 | .758±.022● |
| Enb | .969±.016 | .716±.028● | .988±.012○ | .935±.023● | .792±.033● | .701±.047● | .773±.041● | .769±.082● |
| Song | .788±.021 | .771±.024● | .692±.038● | .785±.029 | .793±.026 | .755±.036● | .784±.022 | .744±.051● |
| BeLaE | .451±.019 | .423±.021● | .357±.019● | .412±.024● | .449±.016 | .410±.012● | .452±.015 | .341±.012● |
| Voice | .971±.005 | .940±.009● | .938±.006● | .958±.009● | .950±.009● | .932±.015● | .954±.008● | .940±.030● |
| Thyroid | .971±.002 | .961±.002● | .956±.003● | .968±.002● | .965±.002● | .960±.003● | .960±.003● | .939±.061 |
| CoIL2000 | .960±.004 | .874±.005● | .897±.005● | .904±.005● | .938±.003● | .949±.009● | .957±.004● | .900±.006● |
| TIC2000 | .945±.004 | .892±.007● | .875±.006● | .885±.004● | .929±.005● | .936±.006● | .945±.004 | .892±.023● |
| Flickr | .798±.005 | .715±.005● | .675±.006● | .735±.005● | .800±.004 | .748±.007● | .795±.003 | .733±.015● |
| Adult | .728±.004 | .701±.004● | .638±.005● | .679±.004● | .723±.003● | .657±.007● | .725±.003● | .697±.004● |
| Default | .677±.003 | .665±.003● | .587±.004● | .663±.002● | .671±.004● | .625±.015● | .676±.003 | .632±.013● |

*Table 3.* Experimental results (mean±std.) of each MDC approach in terms of Exact Match. In addition, ●/○ indicates whether DCOM is significantly superior/inferior to other compared approaches on each data set with pairwise t-test at 0.05 significance level.

| Data Set | DCOM | BR | CP | DLEM | EDCC | LEFA | PIST | SLEM |
|---|---|---|---|---|---|---|---|---|
| Flare1 | .811±.076 | .821±.069 | .718±.071● | .762±.074● | .818±.076 | .805±.074 | .821±.069 | .410±.318● |
| Oes97 | .069±.052 | .030±.027● | .030±.030● | .060±.046 | .057±.045 | .033±.036● | .060±.048 | .018±.031● |
| Jura | .554±.054 | .329±.104● | .512±.052● | .540±.076 | .382±.099● | .315±.041● | .359±.096● | .549±.062 |
| Oes10 | .097±.046 | .064±.033● | .077±.042 | .094±.043 | .099±.043 | .084±.037 | .102±.038 | .062±.034● |
| Enb | .939±.032 | .431±.055● | .975±.024○ | .870±.045● | .584±.067● | .402±.094● | .546±.083● | .611±.091● |
| Song | .480±.054 | .449±.057● | .343±.047● | .478±.059 | .493±.049 | .429±.071● | .467±.047 | .437±.065 |
| BeLaE | .036±.013 | .028±.009 | .013±.009● | .027±.013● | .033±.012 | .017±.008● | .035±.019 | .006±.004● |
| Voice | .942±.010 | .884±.016● | .878±.010● | .918±.017● | .902±.017● | .872±.021● | .910±.016● | .900±.028● |
| Thyroid | .815±.012 | .743±.014● | .712±.016● | .803±.014● | .770±.014● | .737±.019● | .738±.016● | .698±.184 |
| CoIL2000 | .831±.011 | .515±.011● | .616±.013● | .640±.014● | .747±.015● | .786±.036● | .822±.014● | .609±.019● |
| TIC2000 | .842±.011 | .698±.018● | .665±.010● | .688±.009● | .799±.014● | .819±.016● | .843±.013 | .723±.034● |
| Flickr | .328±.013 | .187±.010● | .158±.008● | .226±.005● | .332±.015 | .246±.010● | .330±.013 | .244±.017● |
| Adult | .303±.009 | .228±.006● | .206±.007● | .239±.008● | .281±.008● | .202±.014● | .288±.006● | .288±.013● |
| Default | .199±.005 | .177±.007● | .124±.006● | .181±.007● | .185±.008● | .134±.018● | .195±.006● | .148±.012● |

embeddings. Multi-class algorithms are used for subsequent classification. PIST (Huang et al., 2024) learns pairwise dimension-specific features to consider both the specific characteristics in each dimension and class dependencies among different dimensions. SLEM (Jia & Zhang, 2021) learns a multi-output regression model within an encoded label space by considering the sparse property. For all compared approaches, the recommended parameter setting in the corresponding literatures are used.

### 4.1.4. IMPLEMENTATION DETAILS

For all neural networks $\mathcal{G}, \mathcal{R}, \mathcal{T}$ and $\{\mathcal{H}_j | 1 \leq j \leq q\}$, we employ the multi-layer perceptron (MLP) with one hidden layer, configured with hidden dimension of 512. The dimen-

sionality of latent variable $Z$ is set as 512. In the overall loss function, i.e., Eq.(17), the trade-off parameter $\alpha$ and $\beta$ are both set as 1 (please refer to detailed discussions on parameter sensitivities in Section 4.3.2). All activation functions are fixed as ReLU followed by a dropout layer (Srivastava et al., 2014) with dropping probability of 0.5. For network optimization, we utilize SGD with a batch size of 512, momentum of 0.9 and weight decay of $10^{-4}$. We only adopt experimental results of the last epoch and the number of epoch is uniformly set as 500 for all data sets. Moreover, for the three unstructured MDC data sets, ResNet-18 (He et al., 2016) pretrained on ImageNet dataset (Deng et al., 2009) is implemented as the feature extractor for the proposed approach as well as all compared approaches.

*Table 4.* Experimental results (mean±std.) of each MDC approach in terms of Sub-Exact Match. In addition, ●/○ indicates whether DCOM is significantly superior/inferior to other compared approaches on each data set with pairwise t-test at 0.05 significance level.

| Data Set | DCOM | BR | CP | DLEM | EDCC | LEFA | PIST | SLEM |
|---|---|---|---|---|---|---|---|---|
| Flare1 | .954±.029 | .951±.035 | .935±.032● | .938±.042 | .954±.037 | .941±.038 | .951±.035 | .658±.332● |
| Oes97 | .123±.068 | .072±.048● | .075±.045● | .111±.064 | .123±.072 | .090±.062● | .102±.059● | .039±.038● |
| Jura | .925±.057 | .844±.056● | .838±.070● | .900±.081 | .850±.041● | .769±.104● | .844±.054● | .841±.075● |
| Oes10 | .218±.060 | .119±.056● | .139±.048● | .191±.070 | .201±.061 | .216±.046 | .208±.058 | .152±.046● |
| Enb | 1.00±.000 | 1.00±.000 | 1.00±.000 | 1.00±.000 | 1.00±.000 | 1.00±.000 | 1.00±.000 | .927±.093● |
| Song | .888±.034 | .868±.030● | .763±.064● | .878±.038 | .887±.045 | .841±.055● | .888±.036 | .818±.066● |
| BeLaE | .151±.027 | .132±.023 | .070±.021● | .136±.030 | .157±.021 | .117±.017● | .160±.024 | .068±.014● |
| Voice | 1.00±.000 | .996±.004● | .998±.003 | .999±.002● | .997±.003● | .992±.011● | .997±.003● | .980±.033 |
| Thyroid | .985±.004 | .983±.004 | .980±.005● | .977±.004● | .983±.003 | .982±.004 | .982±.004 | .896±.213 |
| CoIL2000 | .969±.007 | .873±.015● | .905±.010● | .908±.009● | .948±.007● | .963±.007● | .966±.006 | .903±.009● |
| TIC2000 | .993±.003 | .979±.004● | .961±.007● | .966±.005● | .988±.003● | .989±.004● | .993±.002 | .959±.030● |
| Flickr | .729±.015 | .543±.015● | .483±.010● | .600±.014● | .737±.011 | .627±.021● | .723±.009 | .600±.020● |
| Adult | .689±.007 | .657±.009● | .532±.010● | .610±.006● | .687±.006 | .575±.011● | .693±.007○ | .627±.009● |
| Default | .612±.005 | .590±.008● | .446±.008● | .586±.005● | .604±.007● | .518±.032● | .610±.007 | .531±.022● |

*Table 5.* Experimental results of each MDC approach on the three unstructured MDC data sets (i.e., *BP4D*, *DeepFashion*, and *SEWA*) with best results shown in boldface.

(a) Hamming Score

| Data Set | DCOM | BR | CP | DLEM | EDCC | LEFA | PIST | SLEM |
|---|---|---|---|---|---|---|---|---|
| BP4D | **.785** | .745 | .604 | .744 | .754 | .649 | .728 | .547 |
| DeepFa. | **.785** | .771 | .749 | .764 | .774 | .782 | .780 | .713 |
| SEWA | .580 | .537 | .416 | .474 | .559 | .448 | .435 | **.617** |

(b) Exact Match

| Data Set | DCOM | BR | CP | DLEM | EDCC | LEFA | PIST | SLEM |
|---|---|---|---|---|---|---|---|---|
| BP4D | **.274** | .219 | .195 | .246 | .231 | .117 | .186 | .135 |
| DeepFa. | **.285** | .249 | .257 | .242 | .266 | .262 | .262 | .176 |
| SEWA | .290 | .241 | .220 | .199 | .265 | .178 | .121 | **.369** |

(c) Sub-Exact Match

| Data Set | DCOM | BR | CP | DLEM | EDCC | LEFA | PIST | SLEM |
|---|---|---|---|---|---|---|---|---|
| BP4D | **.537** | .458 | .325 | .458 | .479 | .299 | .422 | .244 |
| DeepFa. | **.616** | .591 | .548 | .568 | .592 | .611 | .616 | .478 |
| SEWA | .590 | .540 | .393 | .464 | .563 | .428 | .422 | **.630** |

## 4.2. Experimental Results

We report the detailed experimental results in Table 2, Table 3, Table 4 and Table 5. For structured datasets, ten-fold cross validation are conducted where the mean metric value as well as the standard derivation are recorded for comparison. Furthermore, pairwise t-test (Demšar, 2006) at 0.05 significance level is conducted to show whether DCOM achieves significantly superior/inferior performance against compared approaches. Accordingly, the resulting win/tie/loss counts are summarized in Table 6. Moreover,

Figure 5, Figure 6 and Figure 7 in Appendix C.4 presents the t-SNE visualizations (van der Maaten & Hinton, 2008) for the original features and latent factors in the first fold of dataset *Voice*, *TIC2000* and *Flickr* w.r.t. the first dimension respectively, which shows the learned latent features capture more compact manifold representations than the original features. According to the reported experimental results, we can make some observations as follows:

- For structure data sets, DCOM significantly outperforms the seven compared approaches in $79.6\%$, $73.5\%$ and $60.2\%$ cases in terms of the three evaluation metric, respectively. For unstructured data sets, DCOM surpasses all compared approaches on data set *BP4D* and *DeepFashion* and ranks only second to SLEM on data set *SEWA*.

- There are only 3 cases where DCOM achieves inferior performance against all the other compared approaches. Two of these cases occur on data set *Enb* when compared with CP. This suggests that the two dimensions of *Enb* are strongly correlated, as evidenced by the superior performance of CP in accurately learning the possible class combinations directly. Consequently, it is more appropriate to study the entire probability space for data set *Enb*.

- For data set *SEWA*, SLEM performs the best, likely due to its consideration on label sparsity. This is particularly relevant given the number of classes in each class space of *SEWA* is exceptionally great (21, 20, 19 corresponding to each dimension, respectively).

*Table 6.* Win/tie/loss counts of pairwise t-test (at 0.05 significance level) between DCOM and each compared approach.

| Evalu. | DCOM against | | | | | | | |
|--------|--------|--------|--------|--------|--------|--------|--------|----------|
| Metric | BR | CP | DLEM | EDCC | LEFA | PIST | SLEM | In Total |
| HS | 13/1/0 | 13/0/1 | 11/3/0 | 9/5/0 | 12/2/0 | 7/7/0 | 13/1/0 | 78/19/1 |
| EM | 12/2/0 | 12/1/1 | 10/4/0 | 8/6/0 | 12/2/0 | 7/7/0 | 11/3/0 | 72/25/1 |
| SEM | 10/4/0 | 12/2/0 | 7/7/0 | 5/9/0 | 10/4/0 | 3/10/1 | 12/2/0 | 59/38/1 |
| In Total | 35/7/0 | 37/3/2 | 28/14/0 | 22/20/0 | 34/8/0 | 17/24/1 | 36/6/0 | 209/82/3 |

*Table 7.* Summary of the Wilcoxon signed-ranks test for DCOM against its variants in terms of each evaluation metric at 0.05 significance level. The $p$-values are shown in the brackets.

| DCOM against | HS | EM | SEM |
|--------------|----|----|----|
| DEV1 | win[1.39e-02] | win[7.69e-03] | tie[5.52e-02] |
| DEV2 | win[1.07e-04] | win[3.36e-03] | win[1.07e-04] |

### 4.3. Further Analysis

#### 4.3.1. ABLATION STUDIES

In this section, we further compare the performance of DCOM with its two degenerated versions on all the seventeen MDC benchmark data sets. The two variants, denoted as DEV1 and DEV2, represent two possible model configurations when discarding the proposed class dependency escaping strategy, i.e. the minimization of Eq.(16).

- DEV1. Predictions are derived solely from the conditional marginal probability. In other words, the first term of $\mathcal{L}_{ce}$ (Eq.(15)) and $\mathcal{L}_{ci}$ (Eq.(16)) are discarded.

- DEV2. Predictions are derived solely from the conditional joint probability. In other words, the second term of $\mathcal{L}_{ce}$ (Eq.(15)) and $\mathcal{L}_{ci}$ (Eq.(16)) are discarded. However, the considered sample space remains $\mathcal{D}'$ as defined in Eq.(10) given that the cardinality of original output space $\mathcal{Y}$ is excessively large.

We conduct *Wilcoxon signed-ranks test* (Demšar, 2006; Jia et al., 2025) at significance level 0.05 to analyze whether DCOM performs statistically better than variant models. Table 7 summarizes the $p$-value statistics on each evaluation metric. Compared with these two variant models, we observe that DCOM achieves statistically superior performance against them in terms of each metric. The only comparable case to DEV1 in terms of Sub-Exact Match is notable. It indicates that a classification approach relying exclusively on conditional marginal probability can achieve satisfactory performance when evaluated by metrics with relatively low demands. Nevertheless, when considering evaluation metrics that require higher accuracy, such as Hamming Score

and Exact Match, a strategy that accounts for both conditional marginal and joint probability is preferable.

#### 4.3.2. PARAMETER SENSITIVITY

We show how the the performance of DCOM fluctuates with different dimensionality of latent variable $Z$, different values of trade-off parameter $\alpha$ and $\beta$ in Figure 2, Figure 3 and Figure 4 of Appendix C.1, respectively. It is demonstrated that DCOM achieves relatively stable performance as the three hyperparameters vary within a broad range. However, it is recommended that the dimensionality of latent variable $Z$ should not be set too low, considering its necessity to carry sufficiently rich feature information.

## 5. Conclusion

In this paper, we propose a novel MDC approach DCOM as a first attempt towards escaping from class dependency modeling. This approach provides an effective method to estimate the discrepancy between the conditional joint probability and the product of conditional marginal probabilities. The effectiveness of the minimization of the proposed instance-based conditional mutual information is rigorously validated via comprehensive experiments conducted on seventeen real-world data sets.

## Acknowledgements

The authors wish to thank the anonymous reviewers for their helpful comments and suggestions. This work was supported by the National Science Foundation of China (62225602, 62306131), the SEU Innovation Capability Enhancement Plan for Doctoral Students (CXIH SEU 25134) and the Big Data Computing Center of Southeast University.

## Impact Statement

This paper presents work whose goal is to advance the field of Machine Learning. There are many potential societal consequences of our work, none which we feel must be specifically highlighted here.

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

## A. Proof of Theorem 3.3

Theorem 3.3 shows that when the joint probability of class variables $p_Y(y_{\diamond 1}, y_{\diamond 2}, \ldots, y_{\diamond q})$ is small enough, the absolute value of integrand $|\mathcal{I}(\boldsymbol{y}_\diamond, \boldsymbol{x}_\diamond, \boldsymbol{z}_\diamond)|$ in Eq.(8) tends to 0 under Assumption 3.4. Now we are at a position to prove Theorem 3.3.

*Proof.* Assumption 3.4 indicates that $p^{(\mathrm{jt})}(\boldsymbol{y}_\diamond, \boldsymbol{x}_\diamond, \boldsymbol{z}_\diamond) - \delta \le p^{(\mathrm{pd})}(\boldsymbol{y}_\diamond, \boldsymbol{x}_\diamond, \boldsymbol{z}_\diamond)$.

Consider two cases:

- When $p^{(\mathrm{jt})}(\boldsymbol{y}_\diamond, \boldsymbol{x}_\diamond, \boldsymbol{z}_\diamond) \le p^{(\mathrm{pd})}(\boldsymbol{y}_\diamond, \boldsymbol{x}_\diamond, \boldsymbol{z}_\diamond)$, we have

$$|\mathcal{I}(\boldsymbol{y}_\diamond, \boldsymbol{x}_\diamond, \boldsymbol{z}_\diamond)| = p^{(\mathrm{jt})} \left| \log \frac{p^{(\mathrm{jt})}}{p^{(\mathrm{pd})}} \right| = p^{(\mathrm{jt})} \log \frac{p^{(\mathrm{pd})}}{p^{(\mathrm{jt})}} \le p^{(\mathrm{jt})} \log \frac{1}{p^{(\mathrm{jt})}} = -p^{(\mathrm{jt})} \log p^{(\mathrm{jt})}. \tag{18}$$

  And

$$\lim_{p^{(\mathrm{jt})} \to 0} -p^{(\mathrm{jt})} \log p^{(\mathrm{jt})} = 0, \tag{19}$$

  With the condition that $p_Y(y_{\diamond 1}, y_{\diamond 2}, \ldots, y_{\diamond q}) \to 0$, we have $p^{(\mathrm{jt})}(\boldsymbol{y}_\diamond, \boldsymbol{x}_\diamond, \boldsymbol{z}_\diamond) = p_{Y,X,Z}(\boldsymbol{y}_\diamond, \boldsymbol{x}_\diamond, \boldsymbol{z}_\diamond) \le \int_{\boldsymbol{x}_\diamond} \int_{\boldsymbol{z}_\diamond} p_{Y,X,Z}(y_{\diamond 1}, \ldots, y_{\diamond q}, \boldsymbol{x}_\diamond, \boldsymbol{z}_\diamond) dz = p_Y(y_{\diamond 1}, y_{\diamond 2}, \ldots, y_{\diamond q})$. Therefore, $|\mathcal{I}(\boldsymbol{y}_\diamond, \boldsymbol{x}_\diamond, \boldsymbol{z}_\diamond)| \to 0$.

- When $p^{(\mathrm{jt})}(\boldsymbol{y}_\diamond, \boldsymbol{x}_\diamond, \boldsymbol{z}_\diamond) - \delta \le p^{(\mathrm{pd})}(\boldsymbol{y}_\diamond, \boldsymbol{x}_\diamond, \boldsymbol{z}_\diamond) < p^{(\mathrm{jt})}(\boldsymbol{y}_\diamond, \boldsymbol{x}_\diamond, \boldsymbol{z}_\diamond)$, we have

$$|\mathcal{I}(\boldsymbol{y}_\diamond, \boldsymbol{x}_\diamond, \boldsymbol{z}_\diamond)| = p^{(\mathrm{jt})} \log \frac{p^{(\mathrm{jt})}}{p^{(\mathrm{pd})}} \le p^{(\mathrm{jt})} \log \frac{p^{(\mathrm{jt})}}{p^{(\mathrm{jt})} - \delta}. \tag{20}$$

  And

$$\lim_{p^{(\mathrm{jt})} \to \delta} \lim_{\delta \to 0} p^{(\mathrm{jt})} \log \frac{p^{(\mathrm{jt})}}{p^{(\mathrm{jt})} - \delta} = \lim_{p^{(\mathrm{jt})} \to \delta} \lim_{\delta \to 0} \frac{\delta p^{(\mathrm{jt})}}{p^{(\mathrm{jt})} - \delta} = 0. \tag{21}$$

  Therefore, $|\mathcal{I}(\boldsymbol{y}_\diamond, \boldsymbol{x}_\diamond, \boldsymbol{z}_\diamond)| \to 0$.

$\square$

## B. Detailed Information about Data Sets

In this paper, all structured datasets with detailed descriptions are available in https://palm.seu.edu.cn/zhangml/Resources.htm#MDC_data.

The unstructured dataset *BP4D* is collected from https://www.cs.binghamton.edu/~lijun/Research/3DFE/3DFE_Analysis.html. The original BP4D-Spontaneous dataset (Zhang et al., 2013; 2014) is a 3D video database of spontaneous facial expressions in a diverse group of young adults. Well-validated emotion inductions were used to elicit expressions of emotion and paralinguistic communication. Frame-level ground-truth for facial actions was obtained using the Facial Action Coding System. Eight tasks were covered with an interview process and a series of activities to elicit eight emotions. As well, the Metadata include manually annotated action units (FACS AU), automatically tracked head pose, and 2D/3D facial landmarks. Given the size of the original BP4D-Spontaneous dataset is too large to be used for training efficiently, we sampled part of frames (16037 images) and kept label combinations as much as possible. The first five dimensions correspond to AU06, AU10, AU12, AU14, and AU17 coded with intensity respectively. The intensity codes themselves are either 0 for absent, 1 for present at the A level, 2 for present at the B level, 3 for present at the C level, 4 for present at the D level, 5 for present at the E level; The sixth dimension corresponds to gender where 0 denotes male and 1 denotes female; The seventh dimension corresponds to tasks where 0-7 represents task 1-8.

The unstructured dataset *DeepFashion* is collected from https://mmlab.ie.cuhk.edu.hk/projects/DeepFashion/AttributePrediction.html. The original DeepFashion (Liu et al., 2016) deals with the Category and Attribute Prediction task in Large-scale Fashion (DeepFashion) Database. Category and Attribute Prediction Benchmark evaluates the performance of clothing category and attribute prediction. This is a large subset of DeepFashion,

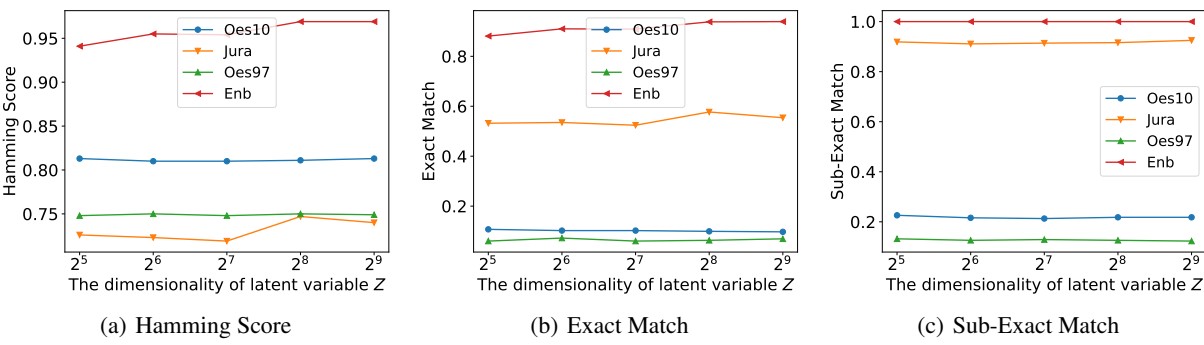

(a) Hamming Score          (b) Exact Match          (c) Sub-Exact Match

*Figure 2.* Performance of DCOM changes as the dimensionality of the latent variable *t* varies in the range of $\{2^5, 2^6, 2^7, 2^8, 2^9\}$.

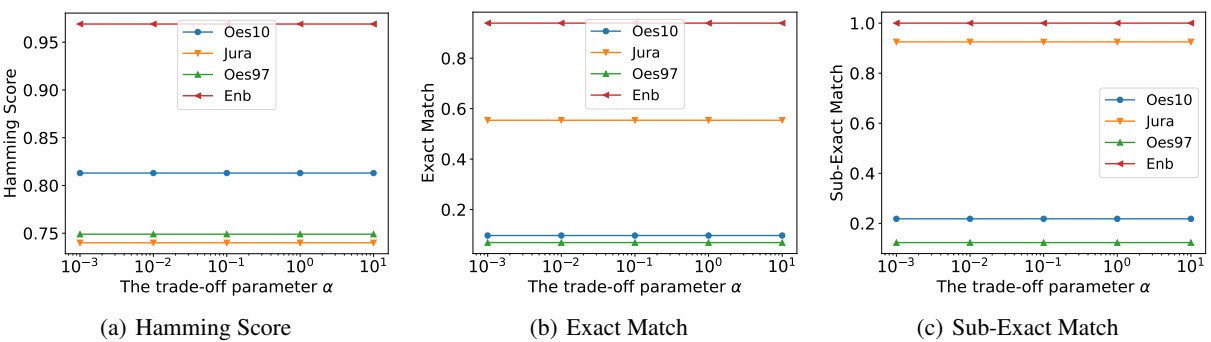

(a) Hamming Score          (b) Exact Match          (c) Sub-Exact Match

*Figure 3.* Performance of DCOM changes as the trade-off parameter $\alpha$ varies in the range of $\{10^{-3}, 10^{-2}, 10^{-1}, 10^0, 10^1\}$.

containing massive descriptive clothing categories and attributes in the wild. We only adopted the fine version of DeepFashion dataset given its multi-dimensional characteristic. The "elasticity" dimension is not mentioned in the original paper. However, in the "list_attr_cloth.txt" in the "Anno_fine" folder of Category and Attribute Prediction Benchmark, we found 3 labels "tight", "loose" and "conventional" which are denoted by "6". We named the sixth dimension as "elasticity".

The unstructured dataset *SEWA* is collected from https://db.sewaproject.eu/. The original SEWA (Kossaifi et al., 2021) is a database of more than 2000 minutes of audio-visual data of 398 people coming from six cultures, 50% female, and uniformly spanning the age range of 18 to 65 years old. The annotation was performed in real-time using a joystick. Specifically, the annotators were asked to push / pull the joystick based on their perception of the subject's level of valence, arousal, or liking / disliking (toward the advert) while being presented the recording. The joystick position (a value between -1000 and 1000) was sampled at 66 Hz and saved into the result file. Given the size of the original SEWA dataset is too large to be used for training efficiently, we sampled part of frames. Finally we kept 19275 images and their corresponding 3 kinds of labels "arousal", "liking" and "valence". We transformed the continuous levels in the range of (-1,1) into discrete labels. For `Arousal` dimension, we transform $(-1, -0.95) \rightarrow 0; [-0.95, -0.85) \rightarrow 1; \ldots; [0.95, 1) \rightarrow 20$; for `Liking` dimension, we transform $(-1, -0.85) \rightarrow 0; [-0.85, -0.75) \rightarrow 1; [-0.75, -0.65) \rightarrow 2; \ldots; [0.95, 1) \rightarrow 19$; for `Valence` dimension, we transform $(-1, -0.85) \rightarrow 0; [-0.85, -0.65) \rightarrow 1; [-0.65, -0.55) \rightarrow 2; [-0.55, -0.45) \rightarrow 3; \ldots; [0.95, 1) \rightarrow 18$. Note that in dimension `Liking` and `Valence`, we merged some intervals of length 0.1 as there are too few examples in the corresponding range.

## C. Additional Tables and Figures

Due to the page limit, we present additional but still important tables and figures in this section.

### C.1. Parameter Sensitivity

Figure 2 shows how the performance of DCOM fluctuates with different dimensionality of latent variable $Z$.

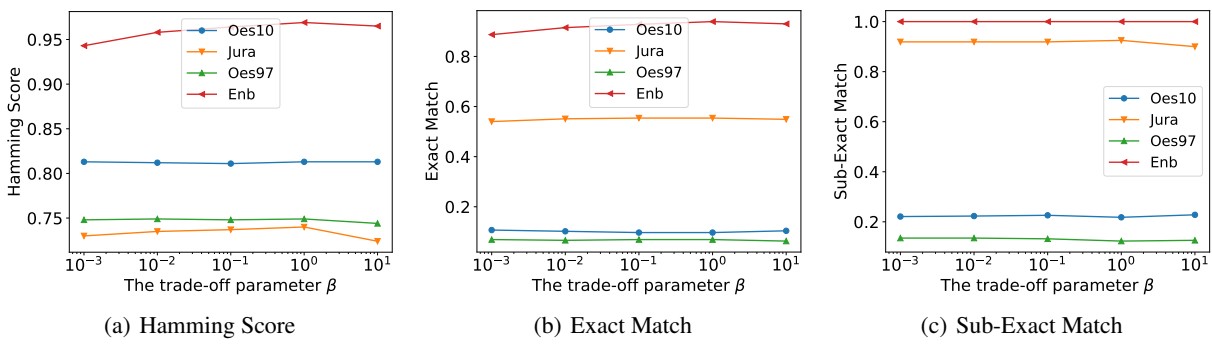

(a) Hamming Score $\qquad$ (b) Exact Match $\qquad$ (c) Sub-Exact Match

*Figure 4.* Performance of DCOM changes as the trade-off parameter $\beta$ varies in the range of $\{10^{-3}, 10^{-2}, 10^{-1}, 10^{0}, 10^{1}\}$.

Figure 3 and Figure 4 shows how the performance of DCOM fluctuates with different values of trade-off parameter $\alpha$ and $\beta$.

## C.2. Notation Table

To facilitate understanding, Table 8 summarizes the notations used in Section 1, Section 2 and Section 3.

## C.3. Pseudo Code

Algorithm 1 presents the pseudo code of the proposed DCOM approach.

---

**Algorithm 1** The DCOM approach

---

**Input**: MDC training set $\mathcal{D}$, an unseen instance $\boldsymbol{x}_*$, the trade-off parameter $\alpha$ and $\beta$
**Output**: Predicted label vector $\hat{\boldsymbol{y}}_*$ for $\boldsymbol{x}_*$

1: Count the number of each class vectors in the training set $\mathcal{D}$ and obtain the set of frequently occurring class vectors $\mathcal{D}'$
2: **repeat**
3:     Randomly sample a batch of examples from $\mathcal{D}$
4:     **for** example $\boldsymbol{x}_i$ in the batch **do**
5:         Corrupt $\boldsymbol{x}_i$ by a random noise $\epsilon$ and obtain $\tilde{\boldsymbol{x}}_i$
6:         Input $\tilde{\boldsymbol{x}}_i$ to the encoding network $\mathcal{G}$ to obtain the corresponding latent vector $\boldsymbol{z}_i$
7:         Input $\boldsymbol{z}_i$ to the classification networks $\{\mathcal{H}_j | q \leq j \leq q\}$ and $\mathcal{T}$ to get the conditional marginal probability $p^{(\mathrm{pd})}(\boldsymbol{y}_\diamond, \boldsymbol{x}_i, \boldsymbol{z}_i)$ and the conditional joint probability $p^{(\mathrm{jt})}(\boldsymbol{y}_\diamond, \boldsymbol{x}_i, \boldsymbol{z}_i)$, respectively
8:         Input $\boldsymbol{z}_i$ to the reconstruction network $\mathcal{R}$ to get the estimated mean vector $\boldsymbol{\mu}_i$ of the posterior probability $p_{X|Z}(\boldsymbol{x}_i|\boldsymbol{z}_i)$
9:         Compute the classification loss $\mathcal{L}_{ce}$ by Eq.(15), the reconstruction loss $\mathcal{L}_{re}$ by Eq.(14) and the loss for minimizing instance-based conditional mutual information by Eq.(16)
10:        Compute the overall loss $\mathcal{L}$ by Eq.(17)
11:        Update the trainable parameters with SGD optimizer
12:     **end for**
13: **until** *Converge*
14: **for** $j = 1$ to $q$ **do**
15:     Feed $\boldsymbol{x}_*$ to the trained encoding network $\mathcal{G}$ and output the latent vector $\boldsymbol{z}_*$
16:     Feed $\boldsymbol{z}_*$ to the classification network $\mathcal{H}_j$ to output the conditional marginal probability $p_{Y_j|X,Z}(y_{\diamond j}|\boldsymbol{x}_*, \boldsymbol{z}_*)$
17:     Get the prediction result on the $j$-th dimension by $\hat{y}_{*j} = \arg\max_{y_{\diamond j} \in C_j} p_{Y_j|X,Z}(y_{\diamond j}|\boldsymbol{x}_*, \boldsymbol{z}_*)$
18: **end for**
19: Return $\hat{\boldsymbol{y}}_* = [\hat{y}_{*1}, \hat{y}_{*2}, \ldots, \hat{y}_{*q}]^\top$

---

## C.4. Visualization of Latent Factors

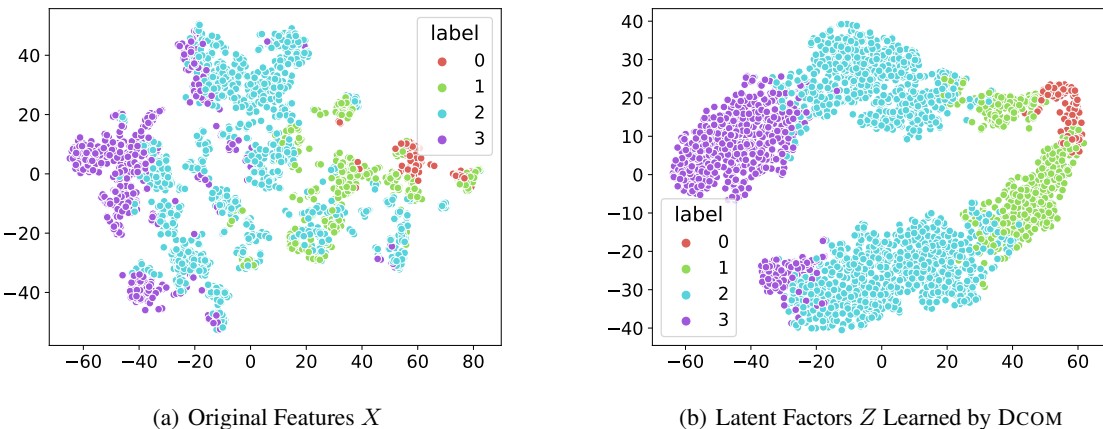

(a) Original Features $X$

(b) Latent Factors $Z$ Learned by DCOM

*Figure 5.* t-SNE Visualization of original feature (left) and latent factors (right) in the first fold of dataset *Voice* w.r.t. the first dimension.

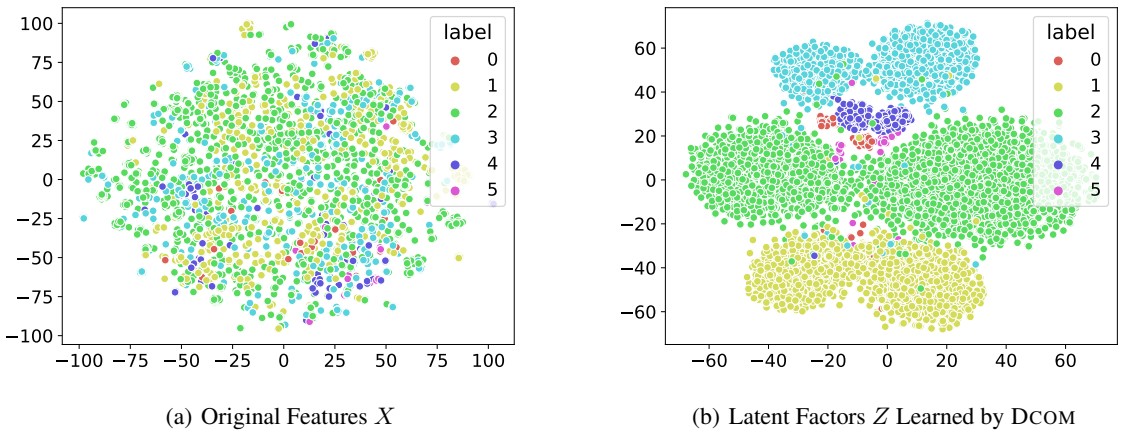

(a) Original Features $X$

(b) Latent Factors $Z$ Learned by DCOM

*Figure 6.* t-SNE Visualization of original feature (left) and latent factors (right) in the first fold of dataset *TIC2000* w.r.t. the first dimension.

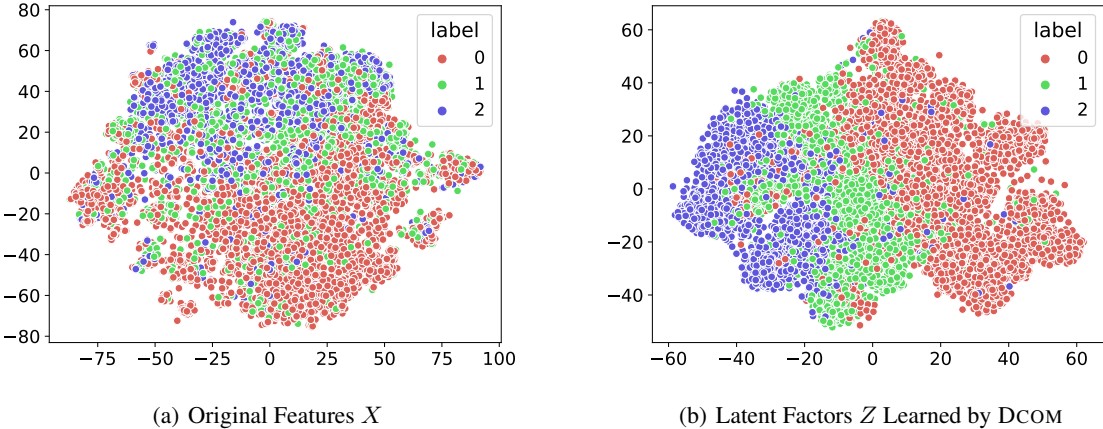

(a) Original Features $X$

(b) Latent Factors $Z$ Learned by DCOM

*Figure 7.* t-SNE Visualization of original feature (left) and latent factors (right) in the first fold of dataset *Flickr* w.r.t. the first dimension.

*Table 8.* Summary of the notations used in our paper.

| Notation | Descriptions |
|---|---|
| $d$ | number of features in input space |
| $q$ | number of class spaces (dimensions) in output space |
| $K_j$ | number of class labels in the $j$-th class space ($1 \leq j \leq q$) |
| $m$ | number of MDC training examples |
| $\mathcal{X}$ | the $d$-dimensional input (feature) space, i.e., $\mathcal{X} = \mathbb{R}^d$ |
| $C_j$ | the $j$-th class space where $C_j = \{c_1^j, c_2^j, \ldots, c_{K_j}^j\}$ ($1 \leq j \leq q$) |
| $c_a^j$ | the $a$-th class label in $C_j$ ($1 \leq a \leq K_j$) |
| $\mathcal{Y}$ | the output space where $\mathcal{Y} = C_1 \times C_2 \times \ldots \times C_q$ |
| $\mathcal{D}$ | the set of MDC samples where $\mathcal{D} = \{(\boldsymbol{x}_i, \boldsymbol{y}_i) | 1 \leq i \leq m\}$ |
| $f$ | the MDC predictive model: $\mathcal{X} \mapsto \mathcal{Y}$ |
| $Y$ | the class variable defined on $\mathcal{Y}$ |
| $Y_j$ | the $j$-th class variable defind on $C_j$ |
| $\boldsymbol{y}_\diamond$ | a randomly sampled value of $Y$ in a probability function |
| $\boldsymbol{y}_i$ | the class vector of the $i$-th example in $\mathcal{D}$ |
| $y_{\diamond j}$ | a randomly sampled value of $Y_j$ in a probability function |
| $y_{ij}$ | the $j$-th element of $\boldsymbol{y}_i$ |
| $X$ | the feature variable defined on $\mathcal{X}$ |
| $\boldsymbol{x}_\diamond$ | a randomly sampled value of $X$ in a probability function |
| $\boldsymbol{x}_i$ | the feature vector of the $i$-th example in $\mathcal{D}$ |
| $\tilde{\boldsymbol{x}}_i$ | the $i$-th corrupted feature vector where $\tilde{\boldsymbol{x}}_i = \boldsymbol{x}_i + \boldsymbol{\epsilon}$ and $\boldsymbol{\epsilon}$ is a random noise vector |
| $\mathcal{D}'$ | the set of frequently occurring class vectors where $\mathcal{D}' = \{\boldsymbol{y}_\diamond | \frac{\#\boldsymbol{y}_\diamond}{m} > c, \boldsymbol{y}_\diamond \in \mathcal{D}\}$ |
| $p^{(\mathrm{jt})}(\boldsymbol{y}_\diamond, \boldsymbol{x}_\diamond, \boldsymbol{z}_\diamond)$ | the conditional joint probability where $p^{(\mathrm{jt})}(\boldsymbol{y}_\diamond, \boldsymbol{x}_\diamond, \boldsymbol{z}_\diamond) = p_{Y,X,Z}(\boldsymbol{y}_\diamond, \boldsymbol{x}_\diamond, \boldsymbol{z}_\diamond)$ |
| $p^{(\mathrm{pd})}(\boldsymbol{y}_\diamond, \boldsymbol{x}_\diamond, \boldsymbol{z}_\diamond)$ | the product of conditional marginal probabilities and joint probability of feature and latent variables where $p^{(\mathrm{pd})}(\boldsymbol{y}_\diamond, \boldsymbol{x}_\diamond, \boldsymbol{z}_\diamond) = \prod_{j=1}^q p_{Y_j|X,Z}(y_{\diamond j}|\boldsymbol{x}_\diamond, \boldsymbol{z}_\diamond) p_{X,Z}(\boldsymbol{x}_\diamond, \boldsymbol{z}_\diamond)$ |
| $\mathcal{I}(\boldsymbol{y}_\diamond, \boldsymbol{x}_\diamond, \boldsymbol{z}_\diamond)$ | the integrant of conditional mutual information where $\mathcal{I}(\boldsymbol{y}_\diamond, \boldsymbol{x}_\diamond, \boldsymbol{z}_\diamond) = p^{(\mathrm{jt})}(\boldsymbol{y}_\diamond, \boldsymbol{x}_\diamond, \boldsymbol{z}_\diamond) \log \frac{p^{(\mathrm{jt})}(\boldsymbol{y}_\diamond, \boldsymbol{x}_\diamond, \boldsymbol{z}_\diamond)}{p^{(\mathrm{pd})}(\boldsymbol{y}_\diamond, \boldsymbol{x}_\diamond, \boldsymbol{z}_\diamond)}$ |
| $\mathcal{G}$ | The encoding network |
| $\mathcal{H}_j$ | The classification network to model conditional marginal probability for the $j$-th dimension |
| $\mathcal{R}$ | The reconstruction network |
| $\mathcal{T}$ | The classification network to model conditional joint probability |
| $Z$ | the latent variable of which the prior is the centered isotropic multivariate Gaussian $p_Z(\boldsymbol{z}_\diamond) = \mathcal{N}(\boldsymbol{0}, \mathbf{I})$ |
| $\boldsymbol{z}_\diamond$ | a randomly sampled value of $Z$ in a probability function |
| $\boldsymbol{z}_i$ | the $i$-th latent vector where $\boldsymbol{z}_i = \mathcal{G}(\tilde{\boldsymbol{x}}_i)$ |
| $\boldsymbol{\mu}_i$ | the mean parameters of the assumptive posterior probability which is a multivariate Gaussian with a diagonal covariance structure $\mathcal{N}(\boldsymbol{\mu}_i, \boldsymbol{\sigma}_i^2 \mathbf{I})$ |
| $\boldsymbol{\sigma}_i$ | the standard deviation parameters of the assumptive posterior probability |
| $\zeta$ | the soft-max function |
| $\mathcal{L}_{ce}$ | the classification loss regarding classification networks $\{\mathcal{H}_j | 1 \leq j \leq q\}$ and $\mathcal{T}$ |
| $\mathcal{L}_{re}$ | the reconstruction loss |
| $\mathcal{L}_{ci}$ | the loss function for minimization of the proposed instance-based conditional mutual information |
| $\mathcal{L}$ | the overall loss function where $\mathcal{L} = \frac{1}{m}(\mathcal{L}_{ce} + \alpha \mathcal{L}_{re} + \beta \mathcal{L}_{ci})$ |

