# OpenReview forum: "Towards Escaping from Class Dependency Modeling for Multi-Dimensional Classification"
_ICML.cc/2025/Conference — ICML 2025 poster_

### Official Review · Reviewer_y3pL · 2025-03-09

**Overall Recommendation:** 1

**Summary:**

The paper proposes an approach to multi-dimensional classification (MDC), named DeCOupling Multi-dimensional classification (DCOM). Different from most MDC methods which explicitly model class dependencies through classifier chains or probabilistic graphical models (PGMs), DCOM captures partial class dependencies by conditioning on original features and latent variables computed from the original features. DCOM is evaluated on a set of benchmark datasets and shows its effectiveness in MDC.

**Claims And Evidence:**

- The assumption of Theorem 3.3 may not be easily satisfied in practice. Can the authors provide examples to demonstrate how it can be held?
- The authors claim that the learned latent factor can capture "critical feature information". However, only empirical results are provided and there are no theoretical results or visualizations for justifying its effectiveness.

**Essential References Not Discussed:**

To my knowledge, no.

**Experimental Designs Or Analyses:**

- The experimental design is mostly sound for validating DCOM's core contributions.

Weaknesses:
- No further analysis or visualization is performed for the learned latent factor.
- The authors claim that DCOM is computationally superior to MDC methods based on learning a graphical model. However, no experiments are performed to support this.

**Methods And Evaluation Criteria:**

The evaluation criteria (HS, EM and SEM) are appropriate for the MDC problem.

**Other Comments Or Suggestions:**

N/A.

**Other Strengths And Weaknesses:**

Strengths:
- The paper is well-written.

Weaknesses:
- The proposed method lacks significant technical novelty.
- The authors did not compare DCOM to MDC methods built upon PGMs. The claim that DCOM is more computationally efficient is not supported by experimental evidence.
- The learned latent variable lacks interpretability.

**Questions For Authors:**

Please see above sections.

**Relation To Broader Scientific Literature:**

DCOM links some aspects of latent variable models to MDC. The effectiveness of latent variable models (e.g. Gaussian mixture models and variational auto-encoders) has been demonstrated in generative modeling tasks in previous studies.

**Theoretical Claims:**

I did not find issues in the derivations in the main text.

---

> ### Author Rebuttal · Authors · 2025-03-30
>
> We want to express our sincere gratitude for your invaluable comments and suggestions. According to your comments in *Claims And Evidence*, *Experimental Designs Or Analyses* and *Other Strengths And Weaknesses*, we summarize the following questions.
> The point-to-point responses are given as follows:
>
> -**Q1: The assumption of Theorem 3.3 may not be easily satisfied in practice. Can the authors provide examples to demonstrate how it can be held?**
>
> Answer for Q1: We sincerely appreciate your insightful question regarding the practical validity of the assumption of Theorem 3.3.
>
> The basic condition of Theorem 3.3 is that the joint probability $p_{Y}(y_{\diamond 1},y_{\diamond 2},\dots,y_{\diamond q}) \to 0$, which can guide selecting high-frequency class combinations when constructing $\mathcal{D}'$ in Eq.(10) (please refer to the answer for Q2 for Reviewer Kvke). Take dataset *Flare1* as an example, the 323 examples present 14 distinct class combinations, where only 3 class combinations correspond to over 10 examples. In other words, the probability of most class combinations is less than 0.03.
>
> As for the low-frequency class combinations involved in Theorem 3.3, we have
> $$p^{\mathrm{(jt)}}(y_{\diamond},x_{\diamond},z_{\diamond})\le \int_{x_{\diamond}}\int_{z_{\diamond}}p^{\mathrm{(jt)}}(y_{\diamond},x_{\diamond},z_{\diamond})dz_{\diamond}\le p_{Y}(y_{\diamond 1},\dots,y_{\diamond q})\to 0,$$
> Accordingly, $p^{\mathrm{(jt)}}-\delta$ also tends to 0, which makes the Assumption 3.4 easily satisfied.
>
> -**Q2: The authors claim that the learned latent factor can capture "critical feature information". However, only empirical results are provided and there are no theoretical results or visualizations for justifying its effectiveness.**
>
> Answer for Q2: Thanks to the comment. Considering that most MDC data sets are tabular data sets, we have not provided visualizations of latent vectors before. For image data sets, we implement pretrained ResNet-18 as the feature extractor and then conduct the proposed approach. So the obtained latent vectors are still tabular.
>
> Figure 1-3 at **anonymous** link <https://anonymous.4open.science/r/DCOM-C8FE/ICML25_authors_response.pdf> present the t-SNE visualizations for the first fold of dataset *Voice*, *TIC2000* and *Flickr*, which show latent factors capture more compact manifold representations than the original features. We will incorporate these additions in the next version.
>
> This dilemma urges us to develop CNN-based MDC approaches to obtain feasible feature maps directly in the future. Thanks again for this insightful comment.
>
> -**Q3: The authors claim that DCOM is computationally superior to MDC methods based on learning a graphical model. However, no experiments are performed to support this.**
>
> Answer for Q3: Thanks to the comment. In Section 2 (*Related Work*), we empirically mention the limitations of existing graphical MDC approaches which are widely recognized in previous works [1][2]. However, computational efficiency is neither the primary focus nor the core contribution of this paper. For comparision with the latest MDC approach *PIST* [3], which shares similar deep learning architecture, we list running times (500 epochs for training) on some data sets as follows which can show our efficiency:
> | Dataset | Time for DCOM (s) | Time for PIST (s) |
> |---|---|---|
> |Flare1|85.97|421.94|
> | Enb|112.94|471.63|
> | Jura|89.88|306.05|
> |Song|167.77|765.10|
> | Oes10|699.84|10219.91|
>
> -**Q4: The proposed method lacks significant technical novelty.**
>
> Answer for Q4: Thanks to the comment. We would like to provide a more comprehensive response addressing this important concern.
> Existing MDC approaches mainly consider modeling class dependencies, where the hardness stems from the typical intercoupling within multiple dimensions. To address this issue, this paper proposes a dependency-free framework and establishes a novel technical route to solve MDC problems. As shown in our experimental results, the proposed approach which only employs conventional MLPs has yielded statistically significant performance improvements over state-of-the-art dependency-modeling baselines.
>
> -**Q5: The authors did not compare DCOM to MDC methods built upon PGMs. The claim that DCOM is more computationally efficient is not supported by experimental evidence.**
>
> Answer for Q5: Please refer to the answer for Q3.
>
> -**Q6: The learned latent variable lacks interpretability.**
>
> Answer for Q6: Please refer to the answer for Q2.
> ***
> [1] B.-B. Jia, M.-L. Zhang. Decomposition-based classifier chains for multi-dimensional classification. IEEE TAI, 2022, 3(2): 176-191
>
> [2] M. Zhu, S. Liu, and J. Jiang, “A hybrid method for learning multidimensional Bayesian network classifiers based on an optimization
> model,” Applied Intelligence, vol. 44, no. 1, pp. 123–148, 2016.
>
> [3] Huang, T., Jia, B.-B., and Zhang,M.-L. Deep multi-dimensional classification with pairwise dimension-specific features. IJCAI'24, pp.  4183–4191.

---

### Official Review · Reviewer_pkvi · 2025-03-11

**Overall Recommendation:** 3

**Summary:**

This submission proposes a feature augmentation approach for multi-dimensional classification (MDC). I think the key motivation is to seek a set of augmented features $\mathbf{Z}$ to fulfill the partial conditional independence (6). In theory, it might work thanks to the notion of conditional independence of random variables. In practice, it might be hard to analyze whether that partial conditional independence is fulfilled or not.

Under the partial conditional independence (6), the joint conditional log-likelihood (5) becomes decomposable as given in (11). Under the assumption of parameter independence, i.e., the parameters of local models used to estimate $\mathcal{H}_j$, $j = 1, \ldots q$, are independent, we can train the local models separately. Section 3.3 details three components of the training loss and introduces the trade-off parameters $\alpha$ and $\beta$.

Experiments are conducted on seventeen data sets to assess the potential advantages of the proposed approach, namely DCOM, compared with other MDC approaches recalled in Section 4.1.3. The empirical evidence suggests that DCOM can provide more promising results, compared to other competitors, and the performance of DCOM seems to be robust to the change of $\alpha$ and $\beta$ and the cardinality of the augmented feature set $\mathbf{Z}$. However, it is not entirely clear to me which type of encoding network has been employed in the experiments.

**Claims And Evidence:**

I think the claims made in the submission are supported by rather clear evidence. However, I would recommend the author(s) to add experiments to assess the robustness of DCOM under the presence of either noisy features or small data sets.

This is because these factors, which can appear in relevant applications, may affect the quality of the augmented feature set $\mathbf{Z}$. I guess under the presence of either noisy features, the augmented feature set $\mathbf{Z}$ may amplify the noisy level of the data. In the case of small data sets, the augmented feature set $\mathbf{Z}$ may amplify the overfitting of the local models $\mathcal{H}_j$, $j = 1, \ldots q$.

**Essential References Not Discussed:**

I think essential references are discussed.

**Experimental Designs Or Analyses:**

On one hand, I think the soundness/validity of any experimental designs and analyses is good and seems to be in favor of the proposed DCOM.

On the other hand, I would recommend the author(s) to add experiments to assess the robustness of DCOM under the presence of either noisy features or small data sets.

**Methods And Evaluation Criteria:**

I think the key assumptions of the proposed DCOM have been either stated in the submission or can be derived from the submission. Commonly used evaluation criteria in the MDC setting have been used in the submission.

**Other Comments Or Suggestions:**

I would use bold capital letters to denote sets of random variables, such as $\mathbf{X}$, $\mathbf{Y}$, and $\mathbf{Z}$ (instead of $X$, $Y$, and $Z$)

**Other Strengths And Weaknesses:**

To my knowledge, DCOM can be seen as a probabilistic graphical model learning approach with multiple latent variables. However, employing parametric models to estimate the local probability distribution may greatly facilitate the optimization of the loss/utility and scalability of the learning phase under suitable assumptions on the structure constraints and (decomposability of) the loss/utility. Therefore, the authors might consider making a connection between DCOM and the literature on that research topic.

**Questions For Authors:**

Q1: I couldn't find details of the encoding network. If I missed some part of the paper, could you give me a pointer? Otherwise, could you add it in the next version?

Q2: Could you make the relations between $log p_{\mathbf{X} | \mathbf{Z}}(\boldsymbol{x}_i | \boldsymbol{z}_i)$ in equation (13) and $\boldsymbol{z}_i$ clearer?

Q3: How do $\mu_{ia}$ and $\delta_{ia}$ related to $\boldsymbol{z}_i$?

Q4: Which kind of reconstruction network has been used in the experiments?

Q5: Have you assessed the robustness of DCOM under the presence of either noisy features or small data sets? Please all refer to "Claims And Evidence" for my detailed comments on this point.

**Relation To Broader Scientific Literature:**

It seems to me that considerable efforts in seeking scalable MDC approaches are based on the notion of conditional independence of random variables, decomposability of the loss/utility and parameter independence. Therefore, this submission may reasonably complement the existing literature on MDC.

**Theoretical Claims:**

I did my best to check the correctness of the theoretical claims. I haven't found any major issue.

---

> ### Author Rebuttal · Authors · 2025-03-30
>
> We want to express our sincere gratitude for your invaluable comments and suggestions. The point-to-point responses are given as follows:
>
> -**Q1: I couldn't find details of the encoding network. If I missed some part of the paper, could you give me a pointer? Otherwise, could you add it in the next version?**
>
> Answer for Q1: Thanks to the comments. Please kindly refer to **​the right column of page 7 (lines 375–384) in Section 4.1.4 (*IMPLEMENTATION DETAILS*)**, the encoding network $\mathcal{G}$ is a **multi-layer perceptron (MLP) with one hidden layer, configured with hidden dimension of 512**. The output dimension, i.e., the dimensionality of latent variable $Z$ is also set as 512. All activation functions are fixed as ReLU followed by a dropout layer with dropping probability of 0.5. We truly hope this clarification can help elucidate details of the encoding network and we will ensure that the implementation details are more prominently highlighted in future revisions.
>
>
> -**Q2: Could you make the relations between $\log p_{X|Z}(\boldsymbol{x}_{i}|\boldsymbol{z}_i)$ in equation (13) and $\boldsymbol{z}_i$ clearer?**
>
> Answer for Q2: We sincerely apologize for the lack of clarity and thank you for identifying this ambiguity. Below we provide a detailed clarification of the relationship between $\log p_{X|Z}(\boldsymbol{x}_i|\boldsymbol{z}_i)$ in Eq.(13) and the latent variable $\boldsymbol{z}_i$:
>
> The term $\log p_{X|Z}(\boldsymbol{x}_i|\boldsymbol{z}_i)$ represents the logarithmic posterior probability of reconstructing input $\boldsymbol{x}_i$ given the latent variable $\boldsymbol{z}_i$, where $\boldsymbol{z}_i$ serves as the **input to reconstruction network $\mathcal{R}$**.
> The reconstruction network $\mathcal{R}$ will then output the mean vector $\boldsymbol{\mu}_i$
> and standard deviation vector $\boldsymbol{\sigma}_i$, i.e., key parameters of the assumed multivariate Gaussian $\mathcal{N}(\boldsymbol{\mu}_i,\boldsymbol{\sigma}_i^2 \mathbf{I})$.
>
> -**Q3: How do $\mu_{ia}$ and $\delta_{ia}$ related to $\boldsymbol{z}_i$?**
>
> Answer for Q3: Please kindly refer to the answer for Q2.
>
> Considering that there is no $\delta_{ia}$ in Eq.(13), we suppose $\delta_{ia}$ in your question is $\sigma_{ia}$.
> $\boldsymbol{z}_i$ serves as the **input to reconstruction network $\mathcal{R}$** and $\mathcal{R}$ will **output** the mean vector $\boldsymbol{\mu}_i$ and standard deviation vector $\boldsymbol{\sigma}_i$.
>
> $\mu_{ia}$ and $\sigma_{ia}$ are the $a$-th element of $\boldsymbol{\mu}_i$ and $\boldsymbol{\sigma}_i$ respectively.
> So they are related directly through the reconstruction network $\mathcal{R}$ as the input and output vectors.
>
> Besides, Eq.(13) aims at inducing the reconstruction loss, i.e., Eq.(14). Thus in the practical algorithm, the reconstruction network $\mathcal{R}$ only needs to output mean parameters $\boldsymbol{\mu}_i$.
>
> -**Q4: Which kind of reconstruction network has been used in the experiments?**
>
> Answer for Q4: Please kindly refer to the answer for Q1. All implementation details are presented in **​the right column of page 7 (lines 375–384) in Section 4.1.4**.  All neural networks ($\mathcal{G}, \mathcal{R}$, $\mathcal{T}$ and {$\mathcal{H}_j|1\le j\le q$}) are employed as the **multi-layer perceptron (MLP) with one hidden layer**, configured with hidden dimension of 512 (except for the feature extractor for image data sets).
>
> -**Q5: Have you assessed the robustness of DCOM under the presence of either noisy features or small data sets? Please all refer to "Claims And Evidence" for my detailed comments on this point.**
>
> Answer for Q5: Thank you for raising this critical question about the robustness of DCOM under noisy features or limited size of data sets. Actually, considering the conditional probability $p(\mathcal{G}(\boldsymbol{x})|\boldsymbol{x})$ is inherently describing a deterministic event, we introduce a minor perturbation on $\boldsymbol{x}$ before inputting it into the encoding network $\mathcal{G}$ (please kindly refer to Eq.(4)). In other words, we have added random noise to the original MDC data sets as Denoising Auto-Encoders [1]. We will make this clearer in the revised version.
>
> As for the small data sets, We have rigorously evaluated DCOM across datasets with varying sizes to assess its generalization capability (please kindly refer to Table 5 in Appendix B). Small data sets such as *Flare1*, *Oes97* and *Jura* only involve less than 400 samples. Though these results demonstrate DCOM's ability to handle limited data, we acknowledge (as correctly noted by you) that the performance advantage is more pronounced on larger datasets. This observation aligns with fundamental learning theory about the data requirements of deep models.
>
> ***
> [1] Vincent, Pascal, Hugo Larochelle, Yoshua Bengio, and Pierre-Antoine Manzagol. “Extracting and Composing Robust Features with Denoising Autoencoders.” In Proceedings of the 25th International Conference on Machine Learning (ICML ’08)

---

### Official Review · Reviewer_tGjW · 2025-03-12

**Overall Recommendation:** 4

**Summary:**

In this paper, the authors propose a new method called DCOM to avoid class dependence modeling in multi-dimensional classification tasks. DCOM introduce an additional estimation of the gap between the joint probability and the product of marginal probabilities. Empirically, the authors verify the effectiveness of DCOM across various multi-dimensional classification tasks.

**Claims And Evidence:**

Most of the claims in this paper are convincing with theoretical and empirical evidence. However, I think it would be better to empirically verify that the performance of previous methods is limited by dependence modeling and DCOM successfully escapes from this issue.

**Essential References Not Discussed:**

I think most of essential references are discussed.

**Experimental Designs Or Analyses:**

The experiments focus on the performance in multi-dimensional classification tasks. I think it would be better to introduce additional experiments to empirically verify that the performance of previous methods is limited by dependence modeling and DCOM successfully escapes from this issue.

**Methods And Evaluation Criteria:**

The introduction of an additional estimation is an interesting idea to solve the dependence modeling, and the proposed methods make sense. Besides, the authors verify their method both theoretically and empirically.

**Other Comments Or Suggestions:**

N/A

**Other Strengths And Weaknesses:**

1. The paper is easy to follow and the core idea is straightforward.
2. The theoretical motivation and empirical results cooperate well.
3. The emprical improvements are not marginal, which verifies the effectiveness of DCOM.

**Questions For Authors:**

1. Is it possible to design a strategy and dynamically adjust the coefficient of the three loss terms?
2. Is it possible to evaluate the influence of dependence modeling in multi-dimensional classification tasks?

**Relation To Broader Scientific Literature:**

N/A

**Theoretical Claims:**

The proof of Theorem 3.3 is clear and I do not find errors.

---

> ### Author Rebuttal · Authors · 2025-03-30
>
> We want to express our sincere gratitude for your invaluable comments and suggestions. The point-to-point responses are given as follows:
>
> - **Q1: Is it possible to design a strategy and dynamically adjust the coefficient of the three loss terms?**
>
> A1: Thanks to the comments. It is indeed possible. In our current experiments, it is shown in Figure 3 and Figure 4 of Appendix C.2 that our approach achieves relatively stable performance as the coefficients vary within a broad range. Therefore, the coefficients of the three loss terms are fixed as 1 during training. This static strategy is chosen to simplify the optimization landscape.
>
> As for possible dynamical strategies, we may suggest adopting gradient balancing techniques [1][2] to automatically align gradient magnitudes across loss terms. This is indeed a promising direction that could further enhance our model's adaptability.
>
> - **Q2: Is it possible to evaluate the influence of dependence modeling in multi-dimensional classification tasks?**
>
> A2: Thank you for raising this critical question about evaluating the influence of dependence modeling in MDC. Your observation highlights a fundamental limitation in current methodological practices. Existing MDC approaches mainly evaluate the influence of dependence modeling implicitly through the downstream **classification metrics** in terms of HS, EM and SEM compared to the baseline approach (BR [3], a classic approach which ignores the dependence modeling completely).
>
> Given the indefinability of dependence, explicit and direct evaluation methods for the influence of dependence modeling may be inaccessible. We suggest seeking possible theoretical definitions of dependence from information theory [4] and further exploring appropriate evaluation methods.
>
> ***
> [1] Yu T, Kumar S, Gupta A, et al. Gradient Surgery for Multi-Task Learning[J]. 2020.DOI:10.48550/arXiv.2001.06782.
>
> [2] Chen Z , Badrinarayanan V , Lee C Y ,et al. GradNorm: Gradient Normalization for Adaptive Loss Balancing in Deep Multitask Networks[J].  2017.DOI:10.48550/arXiv.1711.02257.
>
> [3] Zhang, M.-L., Li, Y.-K., Liu, X.-Y., and Geng, X. Binary relevance for multi-label learning: An overview. Frontiers of Computer Science, 12(2):191–202, 2018
>
> [4] Tishby N. The information bottleneck method[J]. 1999. DOI: 10.1145/345508.345578.

---

### Official Review · Reviewer_Kvke · 2025-03-13

**Overall Recommendation:** 3

**Summary:**

This paper mainly focuses on multi-dimensional classification tasks and points out that existing works mainly focus on designing effective class dependency modeeling strategies but fail to solve the intercoupling of multiple classes. To solve this problem, this paper proposes a method, Dcom, to identify a latent factor that encapsulates the most salient and critical feature information.

**Claims And Evidence:**

The claims in the paper are well supported by evidence.

**Essential References Not Discussed:**

N/A

**Experimental Designs Or Analyses:**

The experimental designs and analyses are reasonable.

**Methods And Evaluation Criteria:**

Yes, the proposed method and evaluation criteria make sense.

**Other Comments Or Suggestions:**

N/A

**Other Strengths And Weaknesses:**

Pros:
- The motivation of the paper is good and clear.
- The paper is well written.
- The paper is well supported by both theoretical and empirical results.

Cons:
- I did not notice the obvious weakness of the paper.

**Questions For Authors:**

- Could you please provide more explanations for the assumptions mentioned in Eq. (3)?
- Does Theorem 3.3 mean that the candidate set will be reduced with the mild assumptions (Assumption 3.4)?
- I am curious about whether VLMs/MLLMs can perform such tasks better than the conventional deep learning methods. An intuition here is that the powerful foundation models can directly handle the multimodal data, like the image and text labels in this paper. It would be interesting to discuss this in the paper.

**Relation To Broader Scientific Literature:**

The contributions are good.

**Theoretical Claims:**

I did not carefully check all theoretical claims in the paper, but the theory parts overall look sound.

---

> ### Author Rebuttal · Authors · 2025-03-30
>
> We want to express our sincere gratitude for your invaluable comments and suggestions. The point-to-point responses are given as follows:
>
> - **Question 1: Could you please provide more explanations for the assumptions mentioned in Eq. (3)?**
>
> Answer for Q1: Eq.(3) serves as a sufficient yet non-necessary condition for ensuring the equivalence between Eq.(1) and Eq.(2). Recall that in MDC, each feature variable $X$ is associated with $q$ class variables $Y_j (1\le j\le q)$, forming a composite class variable $Y=(Y_1,\dots,Y_q)$. Therefore, the class variable considered in Eq. (1) is $Y=(Y_1,\dots,Y_q)$, which models the **joint class structure**. However, existing MDC approaches mainly adopt Eq.(2) as the loss function which considers $q$ scalar class variables $Y_j (1\le j\le q)$ independently and aggregates errors across dimensions via summation (i.e., $\sum_{j=1}^{q}$ over the dimension index).
>
> One straightforward way to impose equivalence between Eq.(1) and Eq.(2) is to assume that each summand is equivalent, i.e., Eq.(3), which actually assumes the partial conditional independence among class variables. We use “partial” here because the assumption does not require Eq.(3) to hold universally for all possible values of $Y$ and $X$, but rather only under the observed training distribution (which is finite and empirically sampled). We will make this clearer in the revised version.
>
>
> - **Question 2: Does Theorem 3.3 mean that the candidate set will be reduced with the mild assumptions (Assumption 3.4)?**
>
> Answer for Q2: Yes, precisely. Eq.(9) involves modeling the class space $\mathcal{Y}$, whose cardinality $|\mathcal{Y}|=\prod^{q}_{j=1}K_j$ grows exponentially with the number of dimensions $q$, resulting in a computationally intractable hypothesis space. By focusing on high-frequency class combinations, the full joint space $\mathcal{Y}$ will be reduced to a sparse subset $\mathcal{D}’$ which is computable in practical algorithms. As shown in the following table, the candidate set is reduced dramatically (with the frequency threshold $c$ set as $0.1$% when constructing $\mathcal{D}’$).
> | Dataset | $\|\mathcal{Y}\|$ | $\|\mathcal{D}’\|$ |
> |---|---|---|
> |Adult|490|37|
> |BeLaE|3125|17|
> |CoIL2000|4800|140|
>
> - **Question 3: I am curious about whether VLMs/MLLMs can perform such tasks better than the conventional deep learning methods. An intuition here is that the powerful foundation models can directly handle the multimodal data, like the image and text labels in this paper. It would be interesting to discuss this in the paper.**
>
> Answer for Q3: We sincerely appreciate the insightful suggestion regarding the potential of VLMs and MLLMs for MDC. We fully agree that foundation models like CLIP or GPT-4V could revolutionize MDC by unifying visual and textual reasoning—a direction we are actively exploring in ongoing work.
> However, our current work faces a practical limitation: most existing MDC data sets (except for the exemplar data set *DeepFashion*) represent class labels as numerical indices (0, 1, $\dots$) rather than rich semantic descriptors. This format significantly constrains our ability to leverage the text-image alignment capabilities that make VLMs/MLLMs so powerful.
>
> Besides, we do believe VLMs/MLLMs will likely demonstrate superior performance with more semantically-rich MDC data sets in the future and will incorporate this important discussion in our revised manuscript to better highlight both the current challenges and future opportunities in applying VLMs/MLLMs models to MDC.

---

### Decision · Program_Chairs · 2025-05-01

**Decision:**

Accept (poster)

**Comment:**

This work proposes a new approach for dealing with multi-dimensional classification where one avoids modelling the relations among class variables and instead rely on some assumptions about their structure (or lack of it). The work is built on some interesting ideas, even though the theoretical part is not strong. Some committee members question whether the assumptions are often satisfied in practice (including wrt the joint class probability), and also whether the experimentation is sufficient to convey all the desired points (including choices of parameters and other decision when putting the experimentation together). Those points are indeed unclear to some extent, but one may say that the overall message remains.